# Batch effects removal for microbiome data via conditional quantile regression

Wodan Ling [1], Jiuyao Lu[2], Ni Zhao [2] ✉, Anju Lulla[3], Anna M. Plantinga [4], Weijia Fu[5], Angela Zhang[1,5], Hongjiao Liu [1,5], Hoseung Song[1], Zhigang Li [6], Jun Chen[7], Timothy W. Randolph [1], Wei Li A. Koay[8,9], James R. White[10], Lenore J. Launer [11], Anthony A. Fodor[12], Katie A. Meyer[3] & Michael C. Wu[1,5] ✉

Batch effects in microbiome data arise from differential processing of specimens and can lead to spurious findings and obscure true signals. Strategies designed for genomic data to mitigate batch effects usually fail to address the zero-inflated and over-dispersed microbiome data. Most strategies tailored for microbiome data are restricted to association testing or specialized study designs, failing to allow other analytic goals or general designs. Here, we develop the Conditional Quantile Regression (ConQuR) approach to remove microbiome batch effects using a two-part quantile regression model. ConQuR is a comprehensive method that accommodates the complex distributions of microbial read counts by non-parametric modeling, and it generates batch-removed zero-inflated read counts that can be used in and benefit usual subsequent analyses. We apply ConQuR to simulated and real microbiome datasets and demonstrate its advantages in removing batch effects while preserving the signals of interest.

Advances in 16S rRNA[1] and full metagenome[2] sequencing technologies have enabled large-scale human microbiome profiling studies involving hundreds to thousands of individuals. The large sample sizes of these studies and the rich availability of metadata promise a comprehensive understanding of the role of microorganisms in health and disease. These studies have already revealed associations between bacterial taxa and both diseases and exposures, such as obesity[3], type 2 diabetes[4], bacterial vaginosis[5], antibiotics[6], and environmental pollutants[7]. However, although large-scale studies facilitate more robust and powerful analyses, they are often subject to serious batch

effects—systematic variation in the data originating from differential handling and processing of specimens[8]. Many large studies include samples collected across times or locations and processed in different runs. In a more extreme situation, several studies may be pooled together for integrative analysis, with inter-study heterogeneity introducing even more severe variation. These batch effects pose serious challenges to analysis and can lead to excessive false positive discoveries, obscure true associations between microbes and clinical variables, and hinder prediction modeling and biomarker development. Unfortunately, despite the importance of batch effects,

[1]Public Health Sciences Division, Fred Hutchinson Cancer Center, 1100 Fairview Ave N, 98109 Seattle, USA. [2]Department of Biostatistics, Johns Hopkins Bloomberg School of Public Health, 615 N Wolfe St, 21205 Baltimore, USA. [3]Nutrition Research Institute and Department of Nutrition, University of North Carolina, 500 Laureate Way, 28081 Kannapolis, USA. [4]Department of Mathematics and Statistics, Williams College, 18 Hoxsey St, 01267 Williamstown, USA. [5]Department of Biostatistics, School of Public Health, University of Washington, 1705 NE Pacific St, 98195 Seattle, USA. [6]Department of Biostatistics, College of Public Health & Health Professions, College of Medicine, University of Florida, 2004 Mowry Rd, 32611 Gainesville, USA. [7]Division of Biomedical Statistics and Informatics, Department of Health Sciences Research, Mayo Clinic, 200 First St SW, 55905 Rochester, USA. [8]Children's National Hospital, 111 Michigan Ave NW, 20010 Washington DC, USA. [9]Department of Pediatrics, George Washington University, Ross Hall 2300 Eye St NW, 20037 Washington DC, USA. [10]Resphera Biosciences, 1529 Lancaster St, 21231 Baltimore, USA. [11]Laboratory of Epidemiology and Population Science, NIA, NIH, 7201 Wisconsin Ave, 20814 Bethesda, USA. [12]Department of Bioinformatics and Genomics, University of North Carolina at Charlotte, 9201 University City Blvd, 28223 Charlotte, USA. ✉e-mail: nzhao10@jhu.edu; mcwu@fredhutch.org

relatively little research has been done on mitigating batch effects for microbiome data.

Batch effects are not unique to microbiome data[9], and standard tools have been developed for other genomic technologies, with the most commonly applied approach being ComBat[10]. However, ComBat and related methods that remove genomic batch effects assume continuous, normally distributed outcomes. Extensions for count data exist[11], but even these make restrictive distributional assumptions. Microbiome data are usually highly zero-inflated, over-dispersed, and heterogeneous with complex distributions. Thus, methods from the other contexts cannot adequately address these issues. At the same time, the limited work on batch effects correction tailored for microbiome data[12–14] can only be used for batch adjustment in association testing or require specific types of controls/spike-ins. These approaches fail to enable other common analytic goals such as visualization or to accommodate more general study designs. Recently, MMUPHin[15] extended ComBat to microbiome analysis by considering zero inflation. But ultimately, it assumes the data to be zero-inflated Gaussian, which is only appropriate for certain transformations of relative abundance data (i.e., taxon counts normalized by each sample's library size). Therefore, more flexible approaches are needed.

In this paper, we propose the conditional quantile regression (ConQuR) approach, a comprehensive microbiome batch effects removal tool. Here, batch removal refers to disentangling the batch effects that could otherwise contaminate the signal of key variables and generating batch-free data that are suitable for any subsequent analyses, while batch adjustment means including batch ID as a covariate in testing. Thus, ConQuR works directly on taxonomic read counts and generates corrected read counts that enable all of the usual microbiome analyses (visualization, association analysis, prediction, etc.) with few restrictions. ConQuR assumes that for each microorganism, samples share the same conditional distribution if they have identical intrinsic characteristics (with the same values for key variables and important covariates, e.g., clinical, demographic, genetic, and other features), regardless of in which batch they were processed. This does not mean the samples have identical observed values, but they share the same distribution for that microbe. Then operationally, for each taxon and each sample, ConQuR non-parametrically models the underlying distribution of the observed value, adjusting for key variables and covariates, and removes the batch effects relative to a chosen reference batch.

ConQuR is fundamentally different from quantile normalization, the widely used approach to align gene expression data. ConQuR allows the underlying taxon abundance distribution to differ across taxa and models the conditional distributions dependent on metadata, while quantile normalization assumes all taxa are homogeneous and makes the empirical marginal distribution identical to a reference batch. Moreover, ConQuR is fundamentally different from genomic batch removal methods. Instead of using parametric models, ConQuR uses a composite non-parametric model to correct the entire complex conditional distribution of microbial read counts, robustly and thoroughly, while maintaining the zero-inflated integer nature of microbiome data. In particular, we use quantile regression for counts[16,17] to model the read counts among samples for which the microbe is present, and separately model the presence–absence status of the microbe by logistic regression. Quantile regression is non-parametric and directly models percentiles of the outcome, such as the median and quartiles. ConQuR is therefore robust to microbiome data characteristics and able to correct higher-order batch effects beyond the mean and variance differences. With zeros explicitly modeled by logistic regression, ConQuR can also address batch variation affecting the presence–absence status of microbes.

To systematically evaluate ConQuR, we conduct simulation studies based on a real vaginal microbiome dataset and examine three large microbiome datasets with different types of batch effects. The real data examples include a gut microbiome study of cardiovascular diseases with moderate batch differences between samples sequenced across several runs, an integrated dataset suffering from more substantial "batch" effects as it comprises different HIV gut microbiome studies, and an oral microbiome study in which batch variation is similar in size to the key variable's effect. By visual and numerical comparisons, we demonstrate that ConQuR thoroughly removes the batch effects and preserves the effects of key variables (continuous, binary, and polytomous) in both association testing and prediction. All usual data transformations and analyses can be conducted on the corrected read count data with minimal regard for the batches.

## Results

### Overview of ConQuR
The central objective of ConQuR is to remove batch effects while preserving real signals in associations in either direction (explaining microbiome variability with the key variable, or vice versa). This is done on a taxon-by-taxon and sample-by-sample basis using a two-step procedure (Fig. 1a). First, in the regression-step, we regress out the batch effects using a non-parametric extension of the two-part model[18] for zero-inflated count outcomes. Specifically, a logistic model determines the likelihood of the taxon's presence, and quantile regression models percentiles of the read count distribution given the taxon is present. The explanatory variables include batch ID, key variables, and scientifically relevant covariates. Accordingly, we can robustly estimate the entire original distribution of the taxon count for each sample, and also estimate the batch-free distribution by subtracting the fitted batch effects relative to a chosen reference batch from both the logistic and quantile parts. Note that we fit the two-part model using all samples for a particular taxon, but due to differences in sample characteristics, the conditional distributions are sample-specific. Second, in the matching-step (Fig. 1b), we locate the sample's observed count in the estimated original distribution, and then pick the value at the same percentile in the estimated batch-free distribution as the corrected measurement. We repeat this two-step correction for each sample and then each taxon. A second version, ConQuR-libsize, directly incorporates library size in the two-part model; thus, in the situation where between-batch library size differences are of interest, the corresponding library size variability is preserved. Both versions are described in more detail in the "Methods" section.

The modeling and estimation framework of ConQuR has four advantages. First, as it directly estimates every conditional percentile without specific assumptions, the complex microbial count distribution is robustly and comprehensively captured. It is more reliable (robust and flexible) than a parametric model, such as negative binomial or Gaussian, which requires the read counts to follow a specific shape. Second, the composite model of logistic and quantile regressions allows heterogeneous associations between the zero-inflated, over-dispersed microbial counts and traits, i.e., batch effects do not need to be uniform across the range of the taxon's abundance. Consequently, the batch effects removal is thorough, mitigating mean, variance, and higher-order batch effects. Finally, as the framework handles zero inflation, it calibrates unwanted presence–absence differences among batches, recovering non-zero counts for under-sampled observations and forcing those over-sampled to be zero.

### Evaluation on simulated data
We simulated data based on MOMS-PI[19], a real vaginal microbiome dataset from the integrative Human Microbiome Project[20], available from the HMP2Data package[21]. After pre-processing, the starting data contain 233 taxa from 270 samples. On top of the intrinsic heterogeneity in the starting data, we simulated 2 conditions (Condition 1 vs. 0) and 2 batches (Batch 1 vs. 0) from a joint Bernoulli distribution with $p_{Condition} = 0.5$, $p_{Batch} = 0.5$, and odds ratio (OR) = 1.25. Thus, Condition is confounded by Batch. We then considered 3 scenarios:

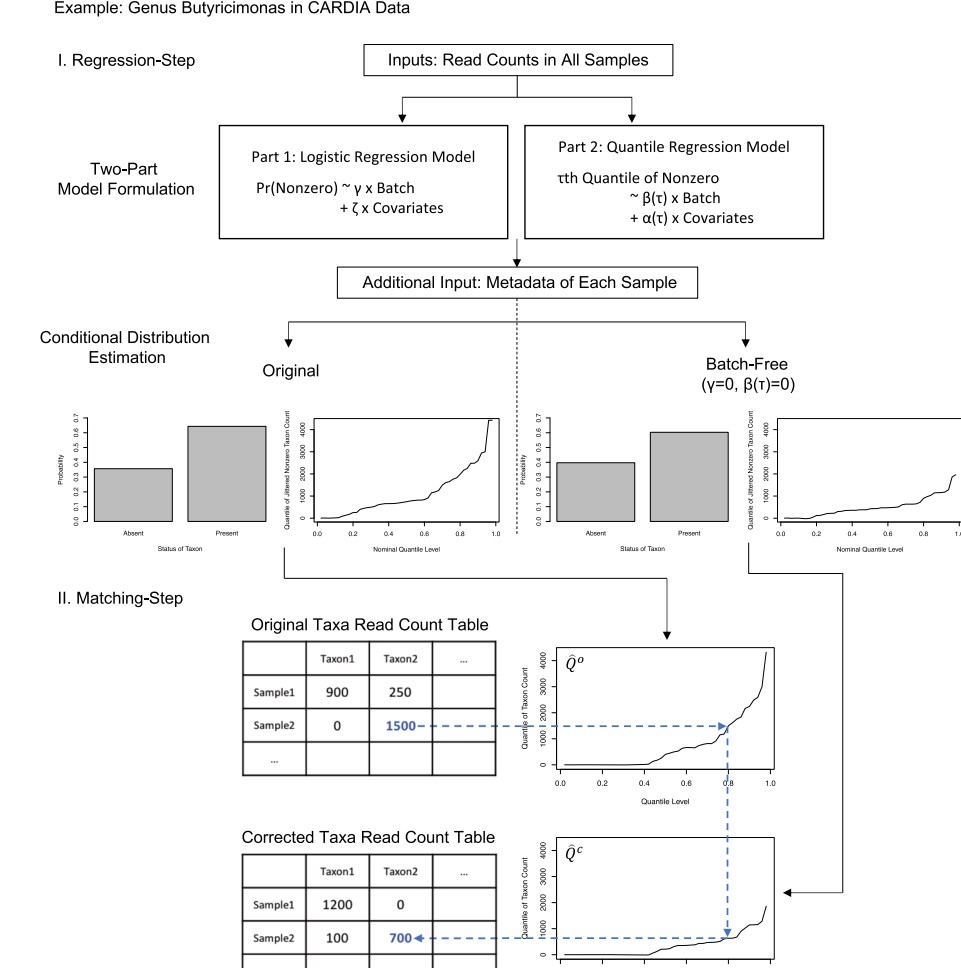

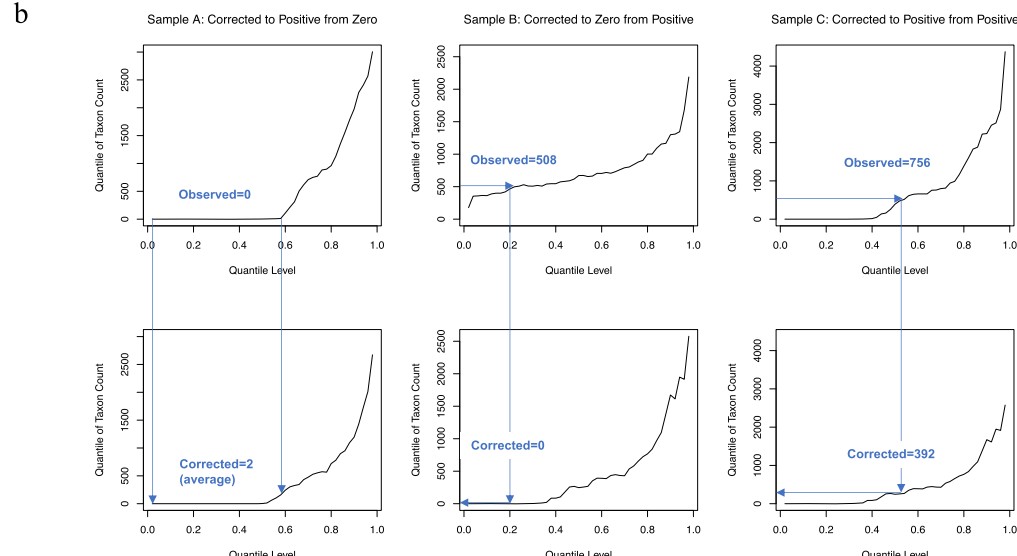

A. Null: Condition fold change (FC) = 16, Batch FC = 1
B. Condition Effect > Batch Effect: Condition FC = 64, Batch FC = 4
C. Condition Effect < Batch Effect: Condition FC = 4, Batch FC = 64

To further challenge ConQuR, we considered Scenarios D, E, and F, which add systematic differences in library size between batches to Scenarios A, B, and C, respectively. Specifically, the probability that a sample belongs to Batch 1 is $p_{Batch} = \frac{1}{1 + \exp(-libsize^s)}$, where $libsize^s$ is the standardized library size (libsize) of each sample in the starting data. Therefore, $p_{Batch}$ is sample-specific and batch effects contain library size variability.

A recurring objective in microbiome studies is association testing for individual taxa. Thus, we chose 20 taxa ranging from the most to the least abundant to be differentially abundant (DA) between

**Fig. 1 | Illustration of the ConQuR algorithm.** Plots are based on real observations of Butyricimonas in the CARDIA study. **a** Two-step procedure. I. regression-step: (1) Use all available samples to fit the two-part quantile regression model; (2) For each sample, estimate the original likelihood of the taxon being present and the original distribution (by estimating a fine grid of percentiles) given the taxon is present. The two parts jointly determines the zero-inflated, over-dispersed conditional quantile function (the inverse of conditional distribution function) of the taxon count $\hat{Q}^o$. In the same manner, estimate the batch-free conditional quantile distribution $\hat{Q}^c$. II. Matching-step: locate the observed read count in $\hat{Q}^o$, and pick the value at the same location of $\hat{Q}^c$ as the corrected read count. Repeat the procedure for each sample and then each taxon. **b** Three scenarios of matching. Left panel: Sample A has a less sparse and less outlying estimated batch-free distribution compared to the original one, so its observed measurement of zero is corrected to be a non-zero number. Middle panel: Sample B has a sparser and more outlying estimated batch-free distribution than the original one, so its observed non-zero count, located at a lower percentile of the original distribution, is corrected to be zero. Right panel: Sample C has a slightly less sparse and less outlying estimated batch-free distribution than the original one, so its observed non-zero count, located at a middle percentile of the original distribution, is corrected to be a smaller non-zero count.

Condition 1 and 0, with the direction of association varying between taxa. Since batch effects affect the entire microbial profile, half of the taxa were set to have increased abundance in Batch 1 (relative to Batch 0) and the other half had decreased abundance in Batch 1.

Next, we mimicked ALDEx2[22] to simulate taxa read counts. Specifically, for sample i, we added 0.5 to its observed count vector in the starting data (to make sure unobserved taxa can also be drawn with minimal probabilities) and used this as the parameter vector to generate relative abundances from a Dirichlet distribution. We then multiplied the simulated relative abundances by $libsize_i$ to generate the initial read counts. Then, if sample i belonged to Condition 1, we divided the initial counts of negatively associated taxa by Condition FC, and multiplied the initial counts of positively associated taxa by Condition $FC'_i$, calculated to maintain $libsize_i$. Finally, if sample i belonged to Batch 1, we divided the counts of taxa with decreased abundance by Batch FC, and multiplied the counts of taxa with increased abundance by Batch $FC'_i$. Additional simulation details, workflow, and data visualization are in Supp. Fig. 3.

We assessed ConQuR from three perspectives: (1) how well the batch effects are removed and condition effects are preserved, (2) the ability of corrected read counts to predict Condition, and (3) the false discovery rate (FDR) and sensitivity of subsequent individual-taxon association analysis for Condition. For (1), we examined the variability of the microbiome data explained by Batch and Condition using PERMANOVA[23] $R^2$. Note that as a measure of multivariate correlation, there is no easy interpretation of PERMANOVA $R^2$; nonetheless, it is a reliable metric to quantify the proportion of variability in microbiome data (assessed by a certain distance matrix) explained by a particular variable. For (2), random forest was chosen to allow for flexible and non-linear modeling. Five-fold cross-validation on the area under the receiver operating characteristic curve (ROC-AUC) was used to evaluate the accuracy. As this analysis merely used prediction accuracy as a complementary metric of evaluation (rather than aiming to evaluate a predictive model), we applied ConQuR to the combined training and testing sets for simplicity. Note that whereas PERMANOVA $R^2$ reflects variability in the taxa explained by Batch and Condition, the ROC-AUC reflects the proportion of Condition explained by taxa. For (3), to evaluate in a general and conservative setting, we used ordinary linear regression of taxon relative abundance on Condition, with FDR controlled by the Benjamini–Hochberg (BH) procedure at $\alpha = 0.05$. Within the taxa table, we computed the observed FDR $\left(\frac{\text{false positives}}{\text{positive calls}}\right)$ and compared it to the nominal value 0.05, and we evaluated the sensitivity $\left(\frac{\text{true positives}}{\text{total positives}} = \frac{\text{true positives}}{20}\right)$.

We repeated the simulation 500 times for each scenario and compared ConQuR with ComBat-seq[11] (designed for RNA-seq count data), MMUPHin[15] (for microbiome count or relative abundance data) and Percentile[12] (for case-control studies with microbiome relative abundance data; we multiplied its output by libsize and rounded to be consistent with the others' outputs) as competing methods.

Figure 2a shows that across all the scenarios, ConQuR reduced the batch variability the most, achieving the lowest Batch PERMANOVA $R^2$ in either Bray-Curtis dissimilarity on the raw count or Euclidean dissimilarity on the corresponding centered log-ratio (CLR)[24,25] transformed relative abundance (Aitchison dissimilarity). At the same time,

it usually preserved the effects of Condition. In terms of the predictive metric, ConQuR also performed the best in maintaining or amplifying the condition signal (Fig. 2b). Collectively, ConQuR outperformed the competing methods in preserving condition effects while thoroughly removing batch effects, enabling more reliable community-level association testing (by PERMANOVA or MiRKAT[26], a generalization of PERMANOVA) and more accurate prediction. Its advantages are most noticeable when batch effects are larger than condition effects (Scenario C and F). ConQuR-libsize demonstrated similar merits.

In the association analysis, ConQuR is the only method that controlled FDR around 0.05 across all the scenarios (Fig. 2c). At the same time, it achieved sensitivity comparable to the other approaches. Percentile appeared to be most powerful, but it could not control FDR and might not be valid. ConQuR-libsize could not control FDR when batch effects were larger than condition effects (Scenario C and F) or batch effects contained library size variability (Scenario E and F). Assessment with nominal FDR cutoffs 0.01 and 0.1 further confirms the findings (Supp. Fig. 4).

To sum up, ConQuR outperforms existing approaches in reducing batch effects and maintaining key signals, especially when batch effects are profound. Moreover, under all circumstances, it controls FDR in subsequent association analysis while achieving satisfactory sensitivity. ConQuR-libsize demonstrates similar or improved performance compared to existing approaches, but it may be inferior to ConQuR in some cases as it ignores the complexity coming from library size variability.

**Application to a single large-scale epidemiology study**

In what follows, we assess ConQuR using real data. We first apply it to a study containing traditional batch variation: samples are collected under one protocol but handled in different batches. The Coronary Artery Risk Development in Young Adults (CARDIA) Study[27] enrolled young adults in 1985–86, with the aim of elucidating the development of cardiovascular disease (CVD) risk factors across adulthood. A variety of clinical risk factors related to CVD were collected, including blood pressure (BP). Basic demographic measures such as age, gender, and race were also collected. At the Year 30 follow-up examination (2015–16), stool samples were collected and processed for DNA extraction and library preparation across four batches. Then, the 16S rRNA marker gene (V3-V4) was sequenced by Illumina technology (MiSeq 2x300) over 7 runs (~96 samples/run), two from each of the first three DNA extraction batches, and the last run from the fourth batch. Thus, at the finest level, data were generated across 7 batches. Following sequencing, forward reads were processed through the DADA2[28] pipeline for quality control and derivation of amplicon sequence variants (ASVs), and taxonomy was assigned using the Silva reference database[29]. The data were aggregated to the genus level, and lineages with zero reads across all samples were excluded.

Batch ID (Batches 0 to 6) indicates in which of the seven sequencing runs each sample was included. Systolic blood pressure (SBP) was the primary variable of interest (SBP > 120 is considered a case for Percentile). Covariates considered for adjustment included gender (Male = 0, Female = 1) and race (White = 0, Black = 1). With

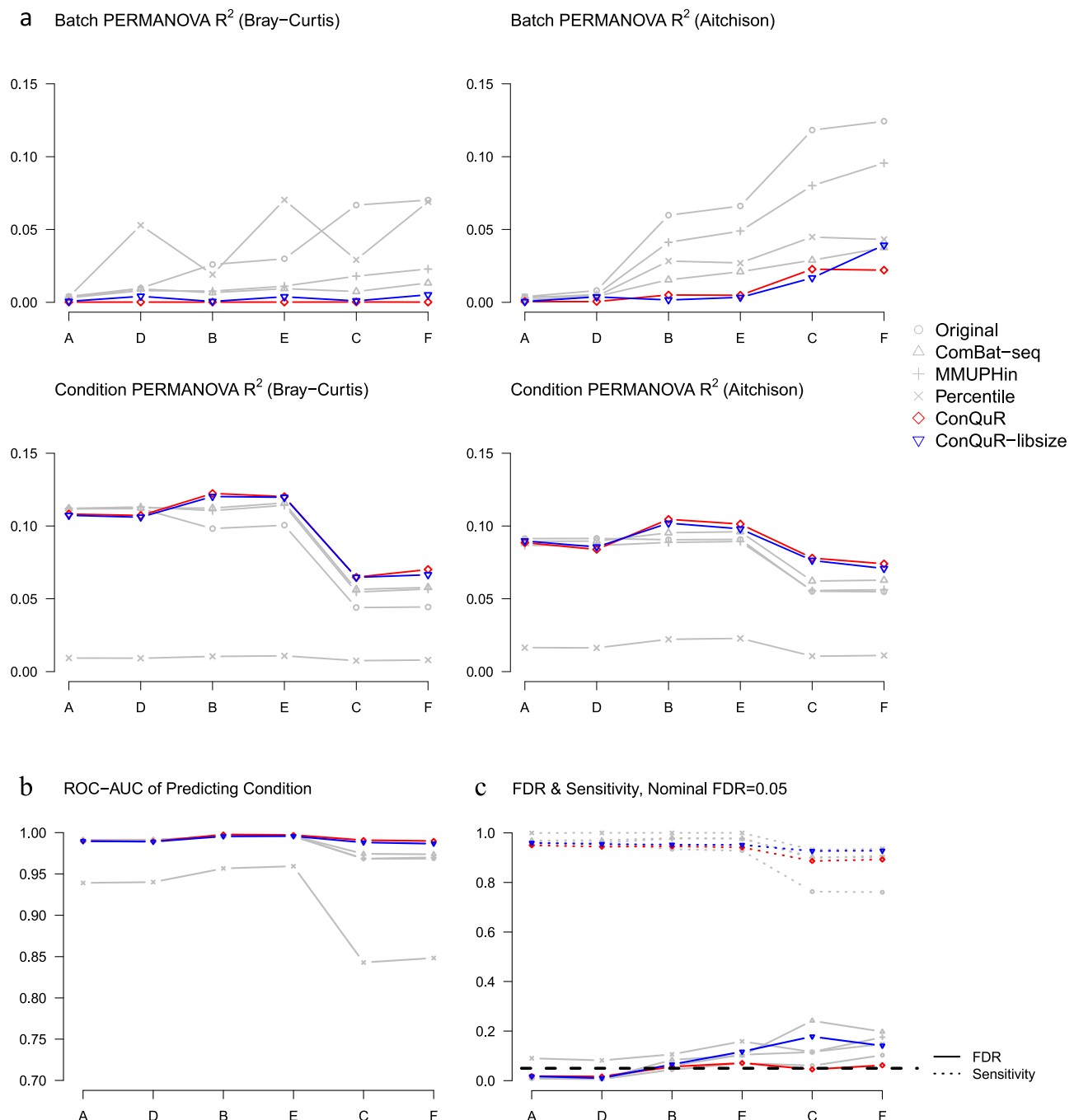

**Fig. 2 | Evaluation on the simulated data.** There are 6 simulation scenarios with 2 conditions and 2 batches, based on the starting data processed from the MOMS-PI study. Simulation scenarios are: A. Condition FC = 16, Batch FC = 1 (Null), B. Condition FC = 64, Batch FC = 4 (Condition Effect > Batch Effect), C. Condition FC = 4, Batch FC = 64 (Condition Effect < Batch Effect), where Condition and Batch are simulated from joint Bernoulli distribution with $p_{Condition} = 0.5$, $p_{Batch} = 0.5$, and OR = 1.25; Scenarios D, E, F are similar to Scenarios A, B, C, respectively, but $p_{Batch} = \frac{1}{1 + \exp(-\text{libsize}^s)}$, making batch variability incorporate library size variability. In the following plots, the scenarios are arranged on the x-axis with the order A, D, B, E, C, F because the two Nulls are allocated together, followed by Condition Effect > Batch Effect, and then Condition Effect < Batch Effect. Color and the name of the corresponding method are shown on the right within the graph. **a** Average proportions of data variability explained by Batch and Condition, quantified by PERMANOVA $R^2$ in either Bray-Curtis or Aitchison dissimilarity. Lower batch variability with preserved or increased condition variability is preferred. **b** Average cross-validated area under the receiver operating characteristic curve (ROC-AUC) of predicting Condition from the taxa read counts via random forest. Higher ROC-AUC indicates a better prediction combining sensitivity and specificity. **c** The average false discovery rate (FDR, solid line) and sensitivity (dashed line) of association analysis between taxa relative abundance and Condition. Approaches with FDR attained around the nominal level 0.05 are valid, and among the valid approaches, higher sensitivity is preferred.

missingness filtered out, the final processed data included 375 genera and 633 samples (Supp. Tab. 1). We aimed to remove the effects of other batches relative to Batch 3, assuming that SBP, gender, and race could jointly describe the conditional distribution for each sample of each taxon's abundance.

We first demonstrated the efficacy of ConQuR through visualization: PCoA plots with colors representing batch IDs. We used Bray-Curtis, Aitchison, and GUniFrac dissimilarities (a compromise between unweighted and weighted UniFrac distances, computed based on relative abundance). As Fig. 3a shows, for all three dissimilarities, the

uncorrected data exhibited significant differences among batches, and ConQuR performed a thorough correction in both the mean (centroids) and dispersion (sizes of ellipses). Specifically, in the raw count scale (by Bray-Curtis dissimilarity), ConQuR centered the means of the seven batches to the same point. As can be seen from the 95% confidence ellipse (an ellipse connects the 95% percentile of points for each batch in the bivariate plot), ConQuR not only equalized the amount of variability across batches but also removed their higher-order effects (angles of the ellipses now are aligned). ConQuR-libsize and the competing methods cannot remove the batch effects as thoroughly as ConQuR. In the relative abundance scale (by Aitchison or GUniFrac dissimilarities), ConQuR also successfully aligned the different batches. However, its advantage over the others was not as substantial as in the raw count scale. This is because ConQuR-libsize and the competing methods either include library size as an offset or work directly on transformations of relative abundance. We also examined ConQuR on common and rare genera separately, showing that compared to competing approaches, ConQuR performed the best on moderate to common taxa (i.e., those present in more than 50% of samples) and demonstrated comparable correction on the rare ones (Supp. Fig. 5).

We then numerically evaluated ConQuR by PERMANOVA[23] $R^2$ and the predictive metric. As Fig. 3b shows, ConQuR induced the largest reduction in the microbiome data variability that can be explained by batch yet maintained the variability that can be explained by SBP, in either the count or relative abundance scale. ComBat-seq showed similar reduction in batch effects in the relative abundance scale but failed to keep the explanatory power of SBP. ConQuR-libsize was not advantageous as ConQuR, but still outperformed the competing methods. Next, we used boxplots to summarize the cross-validated root of mean squared error (RMSE) for predicting SBP from the taxa read counts. ConQuR and ConQuR-libsize systematically lowered the RMSE, amplifying the predictive signal of SBP in the microbial profiles (Fig. 3c).

For the association analysis, at FDR $\alpha = 0.05$, linear regression (adjusting for gender and race) did not find genera associated with SBP in the original, ComBat-seq, or Percentile corrected data. In contrast, Anaerovoracaceae_Family_XIII_UCG-001 (adjusted $p$ value = 0.0012, also identified by MMUPHin) and Hydrogenoanaerobacterium (adjusted $p$ value = 0.0422, also identified by ConQuR-libsize) were detected to be DA in the ConQuR-corrected data. For adolescents, change in Family_XIII_UCG-001's relative abundance is positively related to changes in triglycerides, serum cholesterol, and low-density lipoprotein cholesterol[30], which are factors closely associated with hypertension[31,32]. Also, it is DA between control and coronary artery disease (CAD) patients[33], where the strong link between hypertension and CAD has been shown[34,35]. Hydrogenoanaerobacterium is a crucial contributor to modeling the change of BP in studying the effect of fasting on high BP in metabolic syndrome patients[36]. Supported by the biological findings, we confirm that ConQuR helps to peel off the confounding batch effects, maintain the true signals and lead to meaningful discoveries.

### Application to integration of multiple individual studies

We further consider the performance of ConQuR in the context of vertical data integration where interest is in combining multiple individual studies. We applied it to data from the HIV re-analysis consortium (HIVRC)[37]. Raw 16S rRNA gene sequencing data from distinct studies were processed through a common pipeline—Resphera Insight[38]. Details of data pre-processing and taxonomic assignment are published elsewhere[37]. We focused on the data aggregated to genus level. HIV status (Negative = 0, Positive = 1) was regarded as the primary metadata, while age, gender (Male = 0, Female = 1) and BMI were considered as covariates. Retaining complete cases only, we obtained

the final data that consist of 606 genera for 572 individuals from 10 studies (Supp. Tab. 2) and regarded Study 0 as the reference batch.

Here, the batch effects are between studies and are much more extreme since the studies had varying experimental designs and sequencing protocols (Supp. Tab. 2 of ref. 37). Measured by PERMANOVA $R^2$, the study ID explains 30.39% of the data variability, while the traditional batch effects in CARDIA only contribute 5.66%. We also observed substantial imbalance, sparsity, and heterogeneity in the microbial profiles, as they are unlikely to be fully matched across studies. Comparing Supp. Tab. 2 to Supp. Tab. 1, we see that only 65 out of the 606 genera are present in all studies, while the ratio is 183/375 in CARDIA. Library size ranges also differ greatly across studies, e.g., samples have 185–1000 reads in Study 6, whereas the library size was rarefied to 20,000 reads in Studies 0 and 8. Note that we intentionally kept the samples with minimal library sizes to show ConQuR's capability to handle the outliers. Correcting such heterogenous microbiome data is more challenging than correcting the CARDIA data. The imbalance in metadata (sample sizes and characteristics, Supp. Tab. 2) also adds to the difficulty of batch effects removal.

Visually, we see that ConQuR considerably removed the study variation in the raw count (by Bray-Curtis dissimilarity, Fig. 4a). The means of the 10 studies (centroids) came almost together, and the dispersions and higher-order features (sizes and angles of the confidence ellipses) are much more aligned. In the relative abundance scale (by Aitchison dissimilarity), though ConQuR did not demonstrate perfect correction, it still made the 10 studies substantially more harmonized—brought the means closer and amplified the dispersions of the minimally variable studies, e.g., Study 6, making their variance comparable to the others. We did not conduct the analysis on GUni-Frac dissimilarity because the phylogenetic tree for the pooled HIVRC data was not available to us. ConQuR-libsize performed better than existing methods, but not as well as ConQuR. As before, ConQuR demonstrated more thorough correction on genera with more than 50% prevalence, and was non-inferior on rare genera, compared to the other methods (Supp. Fig. 6).

Numerically, although ConQuR did not make perfect correction of batch effects as on the traditional batch sequencing microbiome data, it maintained its effectiveness in terms of the proportion of unwanted variation eliminated. For the CARDIA data, ConQuR reduced batch effects by 98%, from 5.66% to 0.10%. For the HIVRC data, ConQuR again mitigated 94% of study-to-study variation, from 30.39% to 1.94%, while keeping the importance of HIV status (0.59% vs. 0.57% in the original data, Fig. 4b). On the relative abundance scale, ConQuR still performed the best. Percentile showed slightly more batch reduction on the relative abundance scale, but it failed to preserve the variability explained by HIV status. ConQuR-libsize was the first runner-up in removing batch effects but also did not do well in preserving the key signals. In terms of predicting HIV status, ConQuR boosted the ROC-AUC from 0.75 (from the uncorrected data) to 0.92 and ConQuR-libsize achieved 0.84, while the competing methods failed to enlarge the predictive signal of HIV status in the microbial profiles (Fig. 4c). Overall, ConQuR is robust to different types of batch effects and demonstrated thorough mitigation of batch variation while maintaining signals of interest, even when the batches are highly heterogeneous.

No DA genera between control and HIV+ patient was found in the original data (adjusting for age, gender, and BMI). Acidaminococcus (adjusted $p$ value = 0.0159) was identified in the ConQuR-corrected data only, which has been shown to increase in HIV+ patients[39]. Again, the finding confirms that ConQuR can disentangle signals from the unwanted variation and lead to meaningful discoveries.

### Application to a single study with a large key variable effect size

In both the CARDIA and the HIVRC studies, the batch effects are large compared to the effects of interest (a continuous and a binary variable,

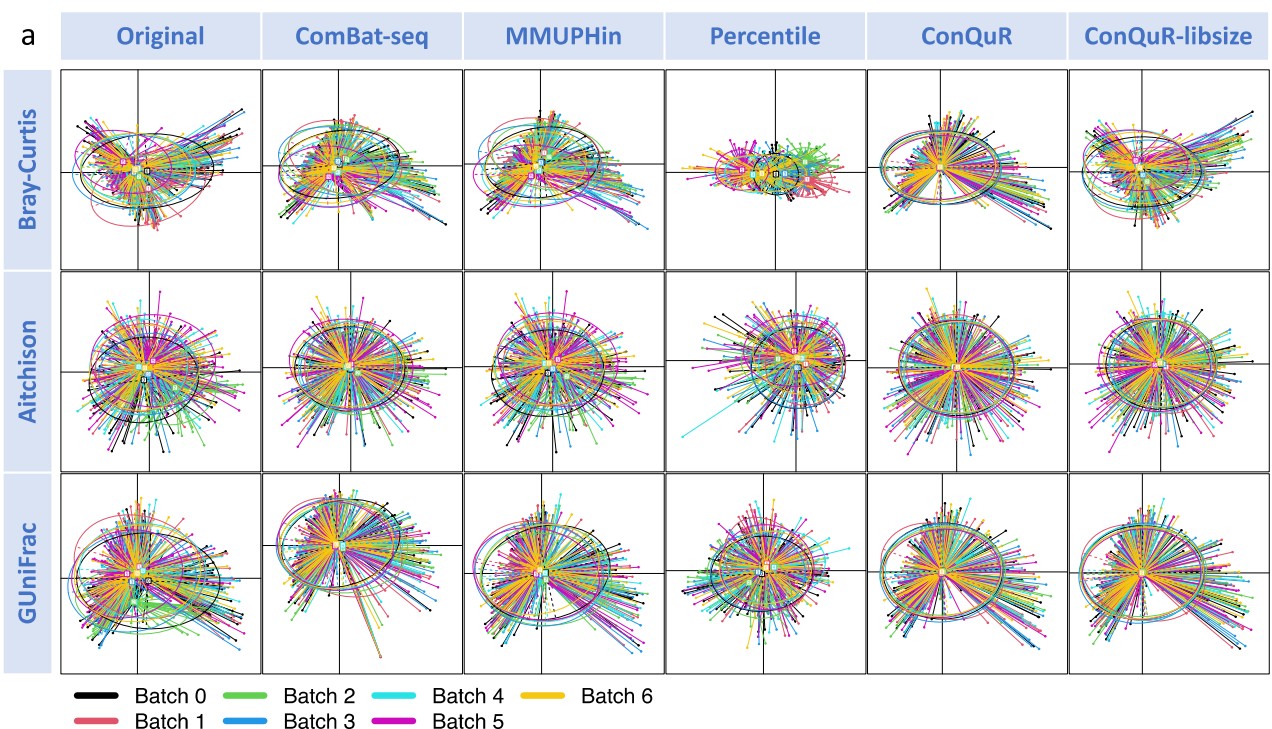

a

| | Original | ComBat-seq | MMUPHin | Percentile | ConQuR | ConQuR-libsize |

Rows: Bray-Curtis, Aitchison, GUniFrac

Batch 0 (black), Batch 1 (red), Batch 2 (green), Batch 3 (blue), Batch 4 (cyan), Batch 5 (magenta), Batch 6 (orange)

b

| PERMANOVA R² | Raw count (Bray-Curtis) | | Relative abundance (Aitchison) | |
|---|---|---|---|---|
| | Batch | SBP | Batch | SBP |
| Original | 0.0566 | 0.0037 | 0.0356 | 0.0035 |
| ComBat-seq | 0.0260 | 0.0032 | 0.0085 | 0.0033 |
| MMUPHin | 0.0311 | 0.0037 | 0.0225 | 0.0034 |
| Percentile | 0.0669 | 0.0026 | 0.0139 | 0.0030 |
| ConQuR | 0.0010 | 0.0038 | 0.0086 | 0.0038 |
| ConQuR-libsize | 0.0234 | 0.0036 | 0.0129 | 0.0038 |

c

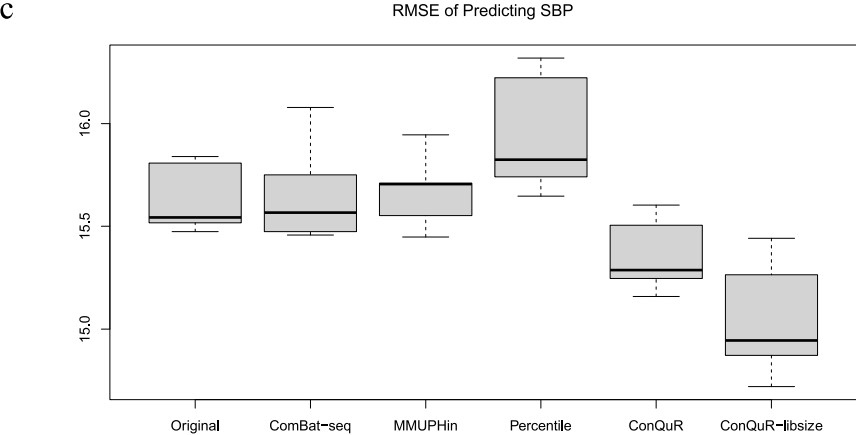

RMSE of Predicting SBP

respectively). We then applied ConQuR to the Men and Women Offering Understanding of Throat HPV (MOUTH) study[40]. In this dataset, the key variable, cigarette smoking (CIG) status, explains comparable amount of data variability as the batch, and has three levels (Never smoker = 0, Former smoker = 1, Current smoker = 2). For the Percentile method, CIG status = 1 or 2 are both considered as cases.

Details about study design, saliva sample collection, and the 16S rRNA sequencing can be found elsewhere[40]. The data were processed through the QIIME2[41] pipeline. We focused on the genus-level data, and considered oral HPV status (Negative = 0, Positive = 1), race (White = 0, Black = 1, Others = 2) and sexual orientation (Heterosexual = 0, Homosexual = 1, Others = 2) as covariates. The final data consists of 247

**Fig. 3 | Evaluation on the CARDIA data. a** PCoA plots clustered by batch ID (corresponding colors are shown at the bottom within the graph), based on Bray-Curtis dissimilarity on raw count data (top panel), Aitchison dissimilarity on the corresponding relative abundance data (middle panel), and GUniFrac dissimilarity on the corresponding relative abundance data (bottom panel). Each point represents a sample and each ellipse represents a batch, with the centroid indicating the mean. As an ellipse connects the 95% percentile of points for each batch, the size of the ellipse indicates the dispersion, and the angle indicates higher-order features of the batch. Better alignment of the ellipses is preferred. **b** Proportions of data variability explained by batch ID and systolic blood pressure (SBP), quantified by PERMANOVA $R^2$ in either Bray-Curtis or Aitchison dissimilarities. Lower variability explained by batch ID with preserved or increased variability explained by SBP is preferred. **c** Cross-validated root of mean squared error (RMSE) of predicting SBP based on the taxa read counts via random forest, where $n = 5$ folds of cross-validation. Lower values indicate stronger predictive signals of SBP in the microbial profiles. Definitions of the boxplot elements: the center line indicates median, the box limits are upper and lower quartiles, whiskers are the 1.5 interquartile range, and points beyond the whiskers are outliers.

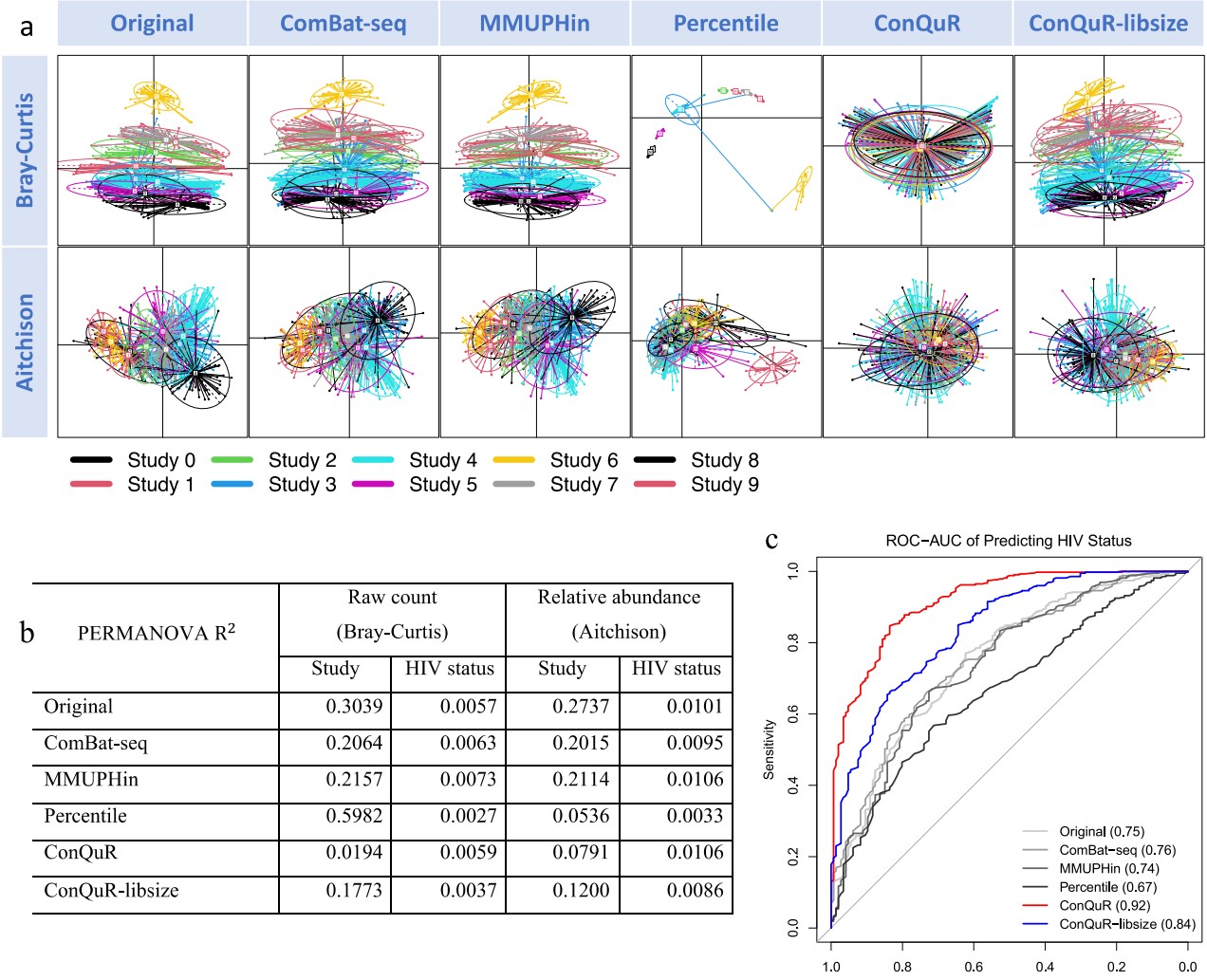

| PERMANOVA $R^2$ | Raw count (Bray-Curtis) | | Relative abundance (Aitchison) | |
|---|---|---|---|---|
| | Study | HIV status | Study | HIV status |
| Original | 0.3039 | 0.0057 | 0.2737 | 0.0101 |
| ComBat-seq | 0.2064 | 0.0063 | 0.2015 | 0.0095 |
| MMUPHin | 0.2157 | 0.0073 | 0.2114 | 0.0106 |
| Percentile | 0.5982 | 0.0027 | 0.0536 | 0.0033 |
| ConQuR | 0.0194 | 0.0059 | 0.0791 | 0.0106 |
| ConQuR-libsize | 0.1773 | 0.0037 | 0.1200 | 0.0086 |

**Fig. 4 | Evaluation on the HIVRC data. a** PCoA plots clustered by study ID (corresponding colors are shown at the bottom within the graph), based on Bray-Curtis dissimilarity on raw count data (top panel) and Aitchison dissimilarity on the corresponding relative abundance data (bottom panel). Each point represents a sample and each ellipse represents a batch, with the centroid indicating the mean. As an ellipse connects the 95% percentile of points for each batch, the size of the ellipse indicates the dispersion, and the angle indicates higher-order features of the batch. Better alignment of the ellipses is preferred. **b** Proportions of data variability explained by study ID and HIV status, quantified by PERMANOVA $R^2$ in either Bray-Curtis or Aitchison dissimilarities. Lower variability explained by study ID with preserved or increased variability explained by HIV status is preferred. **c** Cross-validated area under the receiver operating characteristic curve (ROC-AUC) of predicting HIV status based on the taxa read counts via random forest. Higher ROC-AUC indicates stronger predictive signal of HIV status in the microbial profiles.

genera on 486 individuals from 7 batches (Supp. Tab. 3). We regarded Batch 0 as the reference batch.

Visually, the original MOUTH data does not suffer from serious batch variation. All the batch removal methods further improve the homogeneity of the microbial profiles, while ConQuR did noticeably the best job in unifying the means, dispersions, and higher-order features of the 7 batches, in terms of any dissimilarly (Fig. 5a). Similarly, ConQuR demonstrated improved performance on moderate to

common taxa, and comparable correction on rare taxa, as compared to the other approaches (Supp. Fig. 7).

Numerically, batch ID and CIG status explain comparable proportions of the original data variability (Fig. 5b). ConQuR outperformed all the other methods in mitigating the batch variation and increasing the explanatory power of CIG status, in either raw count or relative abundance scale. ConQuR-libsize was the first runner-up, undoubtedly improved from the existing approaches. The cross-

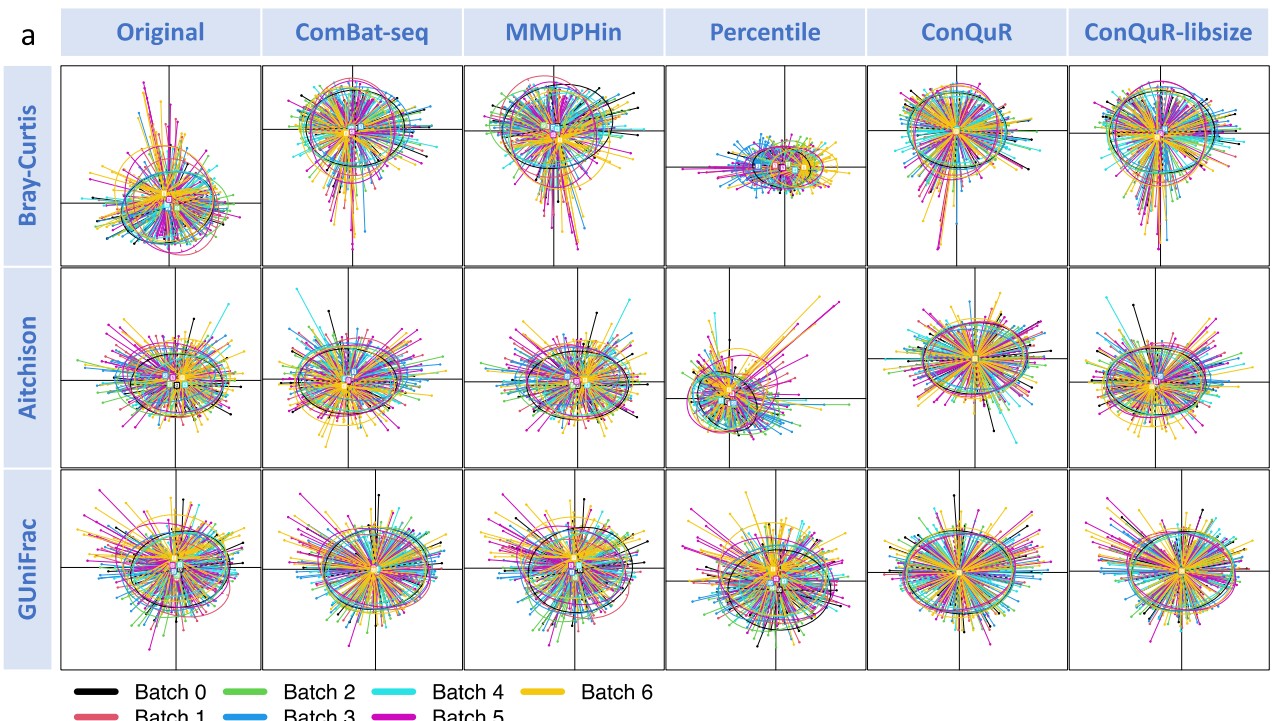

Batch 0
Batch 1
Batch 2
Batch 3
Batch 4
Batch 5
Batch 6

| PERMANOVA R² | Raw count (Bray-Curtis) | | Relative abundance (Aitchison) | |
|---|---|---|---|---|
| | Batch | CIG status | Batch | CIG status |
| Original | 0.0357 | 0.0155 | 0.0454 | 0.0136 |
| ComBat-seq | 0.0238 | 0.0159 | 0.0292 | 0.0139 |
| MMUPHin | 0.0333 | 0.0139 | 0.0434 | 0.0127 |
| Percentile | 0.0489 | 0.0063 | 0.0195 | 0.0070 |
| ConQuR | 0.0006 | 0.0175 | 0.0090 | 0.0143 |
| ConQuR-libsize | 0.0174 | 0.0163 | 0.0113 | 0.0146 |

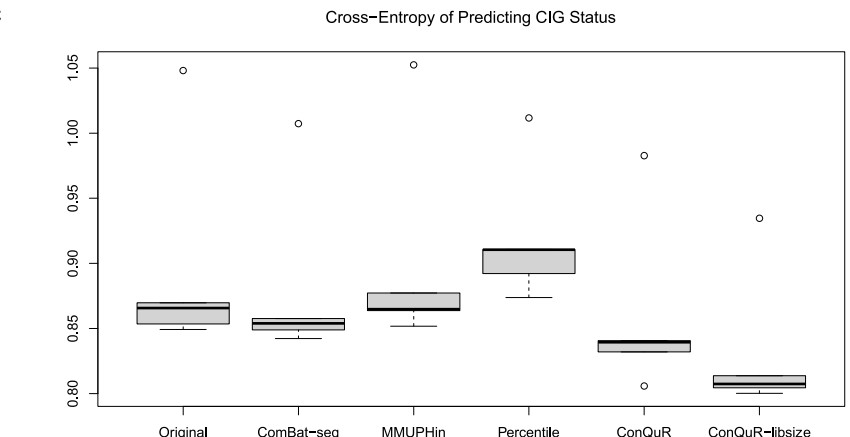

validated cross-entropy of predicting CIG status from the taxa read counts show that ConQuR and ConQuR-libsize were effective in boosting the predictive signal of the polytomous variable in the microbial profiles (Fig. 5c).

No DA genera associated with CIG status were found in the original, ComBat-seq, MMUPHin, Percentile, or ConQuR-libsize corrected data (adjusting for HPV status, race, and sexual orientation). In the ConQuR-corrected data, Coprococcus and 1−68 (Tissierellaceae) (adjusted $p$ values < 0.0001, =0.0071) were identified, where Coprococcus has been shown to be significantly decreased by active smoking[42].

In short, ConQuR demonstrates better performance than the existing methods, for either traditional batch sequencing or integrated data, regardless of the effect size and data type of the key variables.

**Fig. 5 | Evaluation on the MOUTH data. a** PCoA plots clustered by batch ID (corresponding colors are shown at the bottom within the graph), based on Bray-Curtis dissimilarity on raw count data (top panel), Aitchison dissimilarity on the corresponding relative abundance data (middle panel), and GUniFrac dissimilarity on the corresponding relative abundance data (bottom panel). Each point represents a sample, and each ellipse represents a batch with the centroid indicating the mean. As an ellipse connects the 95% percentile of points for each batch, the size of the ellipse indicates the dispersion, and the angle indicates higher-order features of the batch. Better alignment of the ellipses is preferred. **b** Proportions of data variability explained by batch ID and cigarette smoking (CIG) status, quantified by PERMANOVA $R^2$ in either Bray-Curtis or Aitchison dissimilarities. Lower variability explained by batch ID with preserved or increased variability explained by CIG status is preferred. **c** Cross-validated cross-entropy of predicting CIG status based on the taxa read counts via random forest, where $n = 5$ folds of cross-validation. Lower values indicate stronger predictive signals of CIG status in the microbial profiles. Definitions of the boxplot elements: the center line indicates median, the box limits are upper and lower quartiles, whiskers are the 1.5 interquartile range, and points beyond the whiskers are outliers.

## Discussion

Batch effects removal is a challenging task for microbiome data. Approaches designed for genomic data make strong parametric assumptions, which may be inadequate to address the complex distribution of microbiome data. On the other hand, most methods tailored toward microbiome data are restricted to association testing or specialized study designs (such as case-control studies), failing to produce a corrected read count table for other kinds of subsequent analyses under more general designs. The most recently developed method assumes a special parametric distribution, which is only appropriate for certain microbiome data transformations.

We present ConQuR, a robust approach to thoroughly remove unwanted batch variation in microbiome data and generate corrected taxa read count tables. It is based on a two-part quantile regression framework, simultaneously handling the complex read count distribution by quantile regression and the presence–absence status of microbes by logistic regression. ConQuR is most suited for situations in which the microbiome data are processed from highly heterogenous batches and follow irregular distributions. If interest is only in association analysis, it may be sufficient to use existing microbiome batch adjustment approaches. However, ConQuR represents a powerful option—creating batch-removed taxonomic read counts for more general analyses beyond association, including visualization, prediction, etc. We also provide an alternative ConQuR-libsize, incorporating library size as a covariate and an offset in the logistic and quantile regression parts, respectively. The two versions address different needs: ConQuR views the variability from library size as part of the batch effects and mitigates it in the procedure, while ConQuR-libsize explicitly separates it during batch removal and focuses more on the relative measure. We recommend ConQuR as the default as this paper focuses on taxonomic read counts, in which between-batch library size differences are usually considered nuisance variability.

We applied ConQuR to realistic simulated data with two conditions and two batches, as well as to three real microbiome datasets, with the first from a single cardiovascular study with modest batch effects; the second from an integrated analysis—sparser, more irregular, and containing more prominent study variation; and the third from a single oral microbiome study with batch and key variable effects of comparable magnitude. Moreover, the key variables investigated in real examples include continuous, binary, and polytomous variables. Visually and numerically, ConQuR demonstrated rigorous performance in correcting the mean, variance, and higher-order batch effects. At the same time, it preserves the effects of key variables in both association and prediction analyses. Finally, ConQuR facilitates relevant biological discoveries about associations with individual taxa, with the simulation confirming that it helps control FDR and maintain satisfactory sensitivity. The principal advantage of ConQuR lies in its ability to address the complex distributional attributes of microbiome data due to its non-parametric nature, robust to over-dispersion and heterogeneity, and its capability to handle zero inflation. All standard subsequent analyses can benefit from it with minimal regard for batches. Note that we only examined 16S rRNA data in the paper. However, methodologically, ConQuR can also be extended and applied to full metagenome data, which is one of our future directions.

Despite the advantages of ConQuR, it has several limitations which are shared by most existing batch removal procedures. First, comprehensive metadata is required to estimate the conditional distributions of read counts accurately. Second, ConQuR uses the metadata twice, in both the correction and subsequent analyses, theoretically leading to over-optimism in association analysis[43]. However, in practice, this bias is modest relative to the batch effects, and the inclusion of metadata is often helpful for estimating conditional distributions when the taxon is uncommon or imbalanced among batches. Third, ConQuR cannot work if batch completely confounds the key variable. Finally, the performance of all methods depends on the ability to accurately estimate the batch effects, and thus, all methods suffer in the presence of too many small batches (limited information for estimation) and small numbers of sequences/library sizes (poor data quality).

In addition to the shared limitations, our results show that ConQuR is imperfect for low-frequency taxa. This is because quantile regression cannot provide stable estimates with few non-zero read counts, especially at extreme percentiles. In fact, no methods work very well for those very rare taxa. However, ConQuR still performs better than no correction and improves as sample size increases. Consequently, as microbiome profiling studies continue to increase in size, necessarily inducing more batch effects, the performance of ConQuR will only continue to improve.

## Methods

### ConQuR: the conditional quantile regression approach

**Notation.** Suppose we have microbiome data from $n$ samples sequenced in $B$ batches. For each sample, the read counts of $J$ taxa are summarized. We therefore have an $n \times J$ table **Y**, where the entry $Y_{ij}$ denotes the read count of the j-th taxon in the i-th sample. Note that $Y_{ij}$ is a zero-inflated count variable, and we treat it as the outcome in the proposed regression framework. Besides the batch variable $\mathbf{B}_i$, which is expanded as $B - 1$ dummy variables from the batch ID (the one excluded is the reference batch), each sample has a set of characteristics $\mathbf{Z}_i$. $\mathbf{Z}_i$ includes the key variables of interest in subsequent analyses, important biomedical, demographic, genomic and other information based on prior knowledge, and the intercept. Note that we require the key variables to be included, but do not denote them separately from other covariates when presenting the method, because they play similar roles in the batch effects removal procedure. The concatenated $p$-dimension covariates are denoted by $\mathbf{X}_i = (\mathbf{Z}_i^T, \mathbf{B}_i^T)^T$. The proposed method is taxon specific, so we henceforth omit the subscript $j$ for a simpler presentation.

**Regression-step.** One broadly used framework for zero-inflated outcomes is the two-part[18] or hurdle[44] model. It separately models the chance that the taxon is present in a sample and the mean of its abundance given it is present. We employ the same strategy, and first assume that the probability of observing a non-zero $Y_i$, $\pi = P(Y_i > 0 | \mathbf{X}_i)$, follows a logistic regression model,

$$\text{logit}\{P(Y_i > 0 | \mathbf{X}_i)\} = \mathbf{Z}_i^T \boldsymbol{\zeta} + \mathbf{B}_i^T \boldsymbol{\gamma}, \tag{1}$$

where $\boldsymbol{\theta}^L = \left(\boldsymbol{\zeta}^T, \boldsymbol{\gamma}^T\right)^T$ are the true logistic coefficients associated with the covariates and batch variables. Although the presence–absence status of a taxon depends strongly on the sequencing depth, we do not explicitly include the depth in the proposed two-part model, nor do we recommend rarefaction before applying ConQuR. Rather, we assume that batch completely confounds library size (e.g., Batch 1 has mean library size 10,000, Batch 2 has mean library size 20,000, etc.), in which case the presence–absence status depends on batch, and so between-batch library size variability is automatically captured through Model (1). We do not address library size variation within a batch because usually the variation is not substantial and correcting the between-batch variation is our primary goal.

Next, instead of modeling the mean by traditional parametric models, we use linear quantile regression to model the non-zero part, $Y_i | Y_i > 0$. We assume

$$Q_{W_i}(\tau | \mathbf{X}_i, Y_i > 0) = \mathbf{Z}_i^T \boldsymbol{\alpha}(\tau) + \mathbf{B}_i^T \boldsymbol{\beta}(\tau), \tag{2}$$

where $W_i | Y_i > 0 = Y_i | Y_i > 0 + U, U \sim \text{Uniform}(0,1)$, and $\boldsymbol{\theta}^Q(\tau) = (\boldsymbol{\alpha}(\tau)^T, \boldsymbol{\beta}(\tau)^T)^T$ are the true quantile coefficients at the $\tau$-th quantile of $W_i$, which corresponds to a non-zero $Y_i$. Jittering by $U$ is a standard technique to apply quantile regression for counts[17] as it breaks the ties and permits valid estimations and inferences. $\tau$ is a real value between 0 and 1, indicating the quantile level or percentile, e.g., $Q_{W_i}(0.5|\cdot)$ is the conditional median and $Q_{W_i}(0.75|\cdot)$ refers to the conditional third quartile of the jittered non-zero read count. Employing a fine grid of quantile levels $\tau = \left(\frac{1}{k+1}, \ldots, \frac{k}{k+1}\right)$ with a large $k$ (e.g., 5th, ..., 95th percentiles with $k = 19$), we approximately model the conditional quantile function of $W_i | Y_i > 0$. Due to the one-to-one relationship between quantiles of $W_i | Y_i > 0$ and quantiles of $Y_i | Y_i > 0$[17], the conditional quantile function of $Y_i$ given its presence is established.

We fit the two-part quantile regression model (1), (2) to the investigated taxon with all available samples. Combining the results of the two parts based on the fitted models and a sample's metadata, we can estimate the original conditional distribution of the taxon count for that sample. As quantile function is the inverse of distribution function, we estimate the conditional quantile function, which is equivalent to estimating the conditional distribution. First, the estimated probability $\hat{\pi}$ based on the fitted logistic model determines the proportion of zero in the conditional distribution. Specifically, for that sample, all percentiles of the taxon count before the $(1 - \hat{\pi}) \times 100$th percentile are zero. Next, for percentiles after the $(1 - \hat{\pi}) \times 100$th percentile, we squeeze in the estimated conditional quantiles of $Y_i$ given its presence. The combined function on the whole probability spectrum $(0,1)$ is the final estimated conditional quantile function $\hat{Q}^o$, by which the zero-inflated and over-dispersed microbial count distribution is comprehensively revealed. More details of the model and estimation are discussed in the Supplementary Information.

Then, to correct the entire conditional distribution, we regress out batch effects from both the logistic and quantile parts. Specifically, we subtract the estimated effects of batch—$\boldsymbol{\gamma}$, and $\boldsymbol{\beta}(\tau)$ at any percentile, and then combine the two parts in the same manner to obtain the estimated batch-free conditional quantile function $\hat{Q}^c$. Note that by design, $\boldsymbol{\gamma}$ and $\boldsymbol{\beta}(\tau)$ are effects of other batches relative to the reference batch (refer to "Notation" section). Therefore, we eliminate the differences between the sample and those in the reference batch having identical characteristics.

**Matching-step.** With both the original and batch-free conditional distributions in hand, we find the corresponding value of that sample's abundance in the batch-free distribution. Ideally, we can find a unique quantile in $\hat{Q}^o$ equals to the observed count $Y_i$ (find the $\hat{\tau}$ such that $Y_i = \hat{Q}^o(\hat{\tau})$). Then, the value at the same percentile in $\hat{Q}^c$ is the corrected read count $Y_i^c$ ($Y_i^c = \hat{Q}^c(\hat{\tau})$).

Since $Y_i$ is a count variable, there might be multiple quantiles in $\hat{Q}^o$ equal to the observed $Y_i$. It is particularly the case when we adjust zero counts, as microbiome data are zero-inflated. By the strict definition of quantiles—$\tau = P(Y_i \leq y)$, we should locate $Y_i$ at the highest percentile where $\hat{Q}^o$ is less or equal to its observed value. For example, in Fig. 1b (left panel), we need to locate the observed zero at the 58th percentile, the rightmost point of the range where $\hat{Q}^o$ equals zero, and then pick the non-zero count of $\hat{Q}^c$ at the same position as the corrected measurement. However, the estimates, particularly around the zero-positive change point, might not be stable. This is because the estimation of quantile regression is not stable at extreme percentiles. Around the change point, the percentile of non-zero $Y_i$ is approaching the 0th. Therefore, we take the rounded average of all matched quantiles in $\hat{Q}^c$ to obtain a "smoothed" $Y_i^c$, which should be non-zero as well. In this way, we not only allow non-zero values to become zero, but also zero values to become non-zero. Correcting a non-zero value to zero may seem odd since we know the taxon is present. However, it is helpful to keep in mind that zeros in microbiome data may be classified as sampling zeros (due to undersampling) or structural zeros (due to taxon absence), and to understand the introduced zeros as sampling zeros rather than structural zeros. In microbiome studies, there is no way to differentiate between the two kinds of zeros, so we make an assumption that the differences in rate of taxon presence between batches is primarily due to a higher rate of sampling zeros (not structural zeros) in the sparser batches. Instead of recovering the "truth", which will never be fully feasible given the limitations of the data, ConQuR aims to align all batches' distributions, including the presence–absence likelihood.

When the sample size is limited, or the grid of quantile levels is not fine enough, there might be no quantiles in $\hat{Q}^o$ equal to $Y_i$. Then, as quantile functions are left-continuous, we locate $Y_i$ at the percentile with the maximum value smaller than $Y_i$.

After the matching-step, we can correct each sample's observation for the investigated taxon. Repeat the two-step procedure on all taxa, we adjust all entries in the read count table. As both the presence–absence status and values given the presence of all taxa are corrected relative to the reference batch, we observe that, in the corrected table, library sizes of other batches are similar to that in the reference batch and non-zero read counts in other batches follow similar distribution as that of the reference.

**ConQuR-libsize: the alternative of incorporating library size into the regression model.** ConQuR, as described above, is designed for taxa read counts; thus, it considers between-batch library size variability as part of the batch effects and removes it. Another perspective is that since we often treat microbiome abundance as a relative measure, library size should be included in the regression model and the complexity from library size ought to be maintained. Therefore, we provide ConQuR-libsize, of which the two-part model is set as follows,

$$\text{logit}\{P(Y_i > 0 | \mathbf{X}_i)\} = \mathbf{Z}_i^T \boldsymbol{\zeta} + \mathbf{B}_i^T \boldsymbol{\gamma} + \psi \, \text{libsize}_i^s, \tag{3}$$

$$Q_{\log Y_i}(\tau | \mathbf{X}_i, Y_i > 0) = \mathbf{Z}_i^T \boldsymbol{\alpha}(\tau) + \mathbf{B}_i^T \boldsymbol{\beta}(\tau) + \log(\text{libsize}_i), \tag{4}$$

where the standardized libsize is included to model the presence–absence status of the taxon, and in the quantile step, as a standard technique, libsize is treated as an offset to model quantiles of the logarithm of read counts. Note that (2) is equivalent to

$$Q_{\log \frac{Y_i}{\text{libsize}_i}}(\tau | \mathbf{X}_i, Y_i > 0) = \mathbf{Z}_i^T \boldsymbol{\alpha}(\tau) + \mathbf{B}_i^T \boldsymbol{\beta}(\tau), \tag{5}$$

where the outcome is literally the logarithm transformed relative abundance. To estimate the original and batch-free conditional

distributions, we first transform the estimated conditional quantiles of relative abundance given its presence into the count scale (exponentiate, then multiply by libsize), then follow the same procedure as ConQuR.

**Two-layer tuning to achieve the optimal performance.** The choice of reference batch affects the quality of $\hat{Q}^c$ and therefore the performance of ConQuR. Trying several options is recommended if there is no specific preference based on prior knowledge. Note that using the same reference batch across all taxa is crucial to keep the overall structure of microbiome data. Therefore, tuning over potential reference choices is the top layer of the process.

Instead of the standard logistic and quantile regressions, we can use penalized regression or keep the batch ID only in the model, dropping the key variable and covariates, to achieve potentially more stable estimates. Whether the results will benefit from these alternative fitting strategies depends on the specific dataset and frequencies of taxa. Therefore, we suggest a second layer of tuning during which different fitting strategies are used for taxa with different frequencies.

PERMANOVA $R^2$ explained by batch ID is the tuning criterion, with lower values being better. Accordingly, we select the fitting strategy and reference batch that removes batch effects the most. The tuned results are presented in this paper, while visualization of the results by standard ConQuR (using the first batch by numerical or alphabetical order and standard fitting strategy) is included in Supp. Fig. 8.

*Selection of fitting strategies for common to rare taxa*: We start with the second layer of tuning, using a pre-specified reference batch.

For the quantile part, one can use L1-penalized quantile regression[45] with a penalty proportional to sparsity (e.g., $\lambda = \frac{2p}{n_{+vs}}$ or $\frac{2p}{\log(n_{+vs})}$). Like other LASSO methods, this makes the computation feasible when the non-zero counts are fewer than the number of explanatory variables and helps to stabilize the estimates of the conditional quantile functions. A more aggressive alternative is composite quantile regression[46], which forces different quantiles to share the same set of coefficients, except the intercept. It therefore substantially reduces the model complexity, and if the quantiles indeed share similar characteristics (e.g., there are only a few non-zero observations), also improves the estimation accuracy. The option should be used with caution, as its assumption is strong and it is computationally expensive.

For the logistic part, L1-penalized logistic regression can again be applied. However, since there are usually adequate data points for the presence–absence model, this option has limited effect in stabilizing estimation.

The final option is to drop the key variables and covariates, and then use the batch ID exclusively in the regression models for both parts. We call this option simple quantile–quantile matching. In practice, this is essentially the same as matching the empirical quantile functions of each batch to the reference one.

Many factors affect the performance of the standard and alternative fitting strategies, such as the frequencies of taxa, distributional attributes of the read counts (dispersion, heavy tails, or other irregularity), the quality of metadata, etc. Operationally, those options demonstrate different pros and cons for taxa with different frequencies, and the option that is most effective for taxa with a certain range of frequency is data specific. We divide the taxa read count table into sub-tables based on frequency, e.g., sub-Tab. 1 consisting of taxa with prevalence >90%, sub-Tab. 2 consisting of taxa with 80% < prevalence ≤ 90%, etc. For each sub-table, we search for the best fitting strategy that produces the lowest PERMANOVA $R^2$ explained by batch ID. Concatenating the optimally corrected sub-tables, we obtain the overall batch-free microbial profiles. Note that though only local optima (for each sub-table) are determined, this procedure is satisfactory considering the extensive time cost by searching for the global optimum (for the overall taxa read count table).

*Selection of the reference batch*: ConQuR aligns both the presence–absence likelihood and the distribution of counts given the taxon is present to those of the reference batch. Thus, the quality of the reference batch (both taxa counts and sample metadata) is crucial. Note that a large batch or an abundant batch is not necessarily a high-quality batch. For example, if the reference batch is large, consisting of the most samples, but is excessively sparse, counts of other batches will be drawn to zeros as well; if it is abundant, but mostly taking outlying values, corrected measurements of the other batches might be unstably large.

With each potential reference batch, we conduct the second layer tuning and obtain a corresponding optimally corrected taxa read count table. The corrected table with the lowest PERMANOVA $R^2$ is chosen to be the final corrected microbiome data.

**Computation of ConQuR.** The computation of ConQuR is intensive as two conditional distributions (original and batch-free) must be estimated for each sample and each taxon. For a selected fitting strategy, the time depends on sample size $n$ and taxa number $J$, and the scale is approximately $O(nJ)$. Fortunately, the algorithm can be run in parallel because it corrects each taxon separately. In the package, we use two cores to speed it up. For data of similar size as the CARDIA and HIVRC datasets, it will take a PC 15 min and 1.75 GB memory for correction by standard ConQuR. The complexity increases with tuning. It cost a PC 2 hours to fine tune the CARDIA or HIVRC datasets over all possible choices. However, in view of the months or years required for data collection, sample processing and bioinformatics, this one-time computation that increases the quality of subsequent analyses should not be a significant concern.

Also, ConQuR is always computationally feasible, regardless of excessive zeros or outlying non-zero observations. This is because it estimates each location of the conditional distribution, separately and non-parametrically. On the other hand, the algorithm for generating negative binomial realizations of ComBat-seq may fail, because taxa that are highly dispersed or with extreme abundances can lead to extraordinarily large estimates of either the mean or dispersion parameters of negative binomial distribution. In the extreme case, when the continuous covariate, age, was included in the ComBat-seq model when analyzing the CARDIA data, the count generating algorithm failed to converge.

### Description of datasets

CARDIA[27] enrolled young adults in 1985–86, to elucidate the development of CVD risk factors across adulthood. Clinical risk factors related to CVD and demographic measures were collected, including SBP and gender, respectively. Stool samples were collected at the Year 30 follow-up (2015–16) and 16S rRNA gene sequencing data were processed through DADA2[28] pipeline with Silva reference database[29] across 7 runs. CARDIA is used to demonstrate the performance of ConQuR on a single large-scale epidemiology study with moderate batch differences and small effects of interest (SBP, a continuous variable).

HIVRC[37] included multiple individual studies reporting 16S rRNA gene sequences of stool samples from HIV+ patients. The raw sequencing data across studies were processed through Resphera Insight[38]. Details of data pre-processing and taxonomic assignment are published elsewhere[37]. HIVRC is used to show the merit of ConQuR in the context of vertical data integration, where the combined data suffer from more substantial "batch" effects, with small effects of interest (HIV status, a binary variable).

MOUTH[40] reported 16S rRNA gene sequences of saliva samples. Details about study design, sample collection and sequencing can be found elsewhere[40]. MOUTH is used to demonstrate the performance of ConQuR when batch variation is similar in size to the key variable's effect (CIG status, a polytomous variable).

To control the quality of analyses, pre-processing of the CARDIA, HIVRC, and MOUTH data was conducted before batch correction, including aggregating the data to the genus level, removing samples with missing metadata, and removing lineages with zero reads across all samples.

## Reporting summary

Further information on research design is available in the Nature Research Reporting Summary linked to this article.

## Data availability

The MOM-PI data are provided in the Bioconductor R package, HMP2Data. The CARDIA data can only be shared upon request due to the CARDIA Study Publications Policy. A request can be made by submitting a CARDIA Data Set Request-Intent to Analyze Form to the CARDIA Coordinating Center, University of Alabama at Birmingham, at https://www.cardia.dopm.uab.edu/publications-2/publications-documents. The reference database, Silva, can be found at https://www.arb-silva.de/. The integrated HIVRC data are available in Synapse under accession code syn18406854. The MOUTH data are available in Synapse under accession code syn26529406.

## Code availability

The R package ConQuR[47] is available at https://github.com/wdl2459/ConQuR in formats appropriate for Macintosh, Windows, or Linux systems. A vignette demonstrating use of the package (a full analysis pipeline, including the standard fitting strategy, penalized fitting strategy, the fine-tuned result, and investigations on the original and batch-removed taxa read count tables) is included (https://wdl2459.github.io/ConQuR/ConQuR.Vignette.html).

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

## Acknowledgements

This work was supported in part by R01 GM129512 (M.C.W.) and R01 HL155417 (M.C.W.). The Coronary Artery Risk Development in Young Adults Study (CARDIA) is supported by contracts HHSN268201800003I, HHSN268201800004I, HHSN268201800005I, HHSN268201800006I, and HHSN268201800007I from the National Heart, Lung, and Blood Institute (NHLBI). The HIVRC data used in this study are from work that was supported by the HIV Microbiome Re-analysis Consortium. The authors thank Drs. Susan A. Tuddenham and Cynthia L. Sears for their support and review of the manuscript, also thank Dr. Khalil G. Ghanem and all members on the HIV Microbiome Re-analysis Consortium for collecting and processing the HIVRC dataset. Finally, we thank Dr. Ying Zhou for helping generate simulation plots.

## Author contributions

W.L. developed the method, analyzed the data, and wrote the manuscript. J.L., N.Z., A.M.P., W.F., A.Z., H.L., H.S., Z.L., J.C., and T.R. contributed to the conception, methodological developments, and presentation. A.L., W.L.A.K., J.R.W., and L.J.L. contributed to the dataset generation and manuscript development with technical input. A.A.F. and K.A.M. contributed to dataset/variable generation, conception, methodological developments, and presentation. M.C.W. conceived the study and critically reviewed the manuscript. All authors read and approved the final manuscript.

## Competing interests

The authors declare no competing interests.
