## [Peer Review File · Nature Communications]

REVIEWER COMMENTS

Reviewer #1 (Remarks to the Author):

In Ling et al., the authors proposed the quantile regression-based method ConQuR, for batch-correction of microbiome count observations. The method is composed of two components targeting the presence/absence and the non-zero part of microbiome count distributions. Additional considerations include model tuning for the reference batch, and a piecewise estimation strategy to prevent instability at the left tail of the non-zero distribution. The authors showcase the method's performance, by applying it to two real-world datasets: one composed of one study of different sequencing batches, the other composed of different cohorts with more prominent "batch" effects. In either case ConQuR demonstrated excellent performance in both reducing overall community profile batch effects, as well as enhancing biological signals post batch correction. It also outperformed existing RNA-Seq batch correction method (ComBat).

I find the proposed quantile regression method elegantly designed, and is well-suited for microbiome data batch correction. The details of the method are clearly explained and easy to follow. The real-world dataset results to showcase ConQuR's performance are convincing and well-presented. Still, I do have a few major-to-moderate comments. These mostly concern the model's design philosophy/interpretations.

Major

1. I wonder about the appropriateness of omitting library size in the model design. The authors suggest that library size effect can be considered as part of the batch-specific effect. This argument applies to sequencing runs as batches, but not necessarily to experimental batches, or study effects. More importantly, removing library size in modelling count data might lead to misleading interpretations. Imagine the toy case of two batches, no other covariate effects. In Batch 1 microbe A has on average 1000 reads out of 3000 total. In Batch 2 microbe A has ~ 2000 reads out of 10000 total. In this example, the count-based batch effect in Batch 2 vs. 1 is positive, whereas the batch effect in relative abundance in Batch 2 vs. 1 is negative.

By extending this toy case to include also biological variables, one can come up with (perhaps contrived) cases where, without modelling for library size, the proposed method might mistakenly estimate both biological and batch signals. The upshot being, for sequencing-based microbiome observations, the per-sample library size is considered an artificial technical signal, and thus often

explicitly accounted for in modelling with e.g. offset terms or as a covariate, to avoid confounding effects.

Thus, I wonder why the authors decided not to consider this variable in their modelling. It was pointed out in their evaluations (Figures 2/3) that ConQuR performed better in count-based than for relative abundance-based evaluations. Maybe accounting for library size can help improve this? On the flip note, the evaluations for e.g. G-UniFrac and prediction modelling still showcased ConQuR's superiority over ComBat, so empirically ignoring library size seemed acceptable for these two datasets.

2. For the presence-absence logistic modelling, the same library size concern applies. But additionally, I wonder about the appropriateness of converting non-zero counts to zero values during batch correction, as this is a potential outcome of ConQuR (Figure 1b Sample B). Imagine the application case of correcting for sequencing batch effects. For a microbe to have substantial non-zero counts in certain batches, one would reason that that such observations indicate actual biological presence of the microbe, and are not mere sequencing technical effects. Thus, correcting some of these non-zero counts to zero values seems difficult to justify to me. The counterpart – imputing zero values to non-zero values seems more acceptable, as the microbe might not have been detected in certain batches due to e.g. under-detection.

Moderate

1. First, I'd like to complement the authors on their considerations regarding model tuning, and the three-piece estimation strategy. It showed great care in designing a method that's well-tailored to microbiome data applications.

2. It seems that the regression coefficients for non-zero counts, $\alpha(\tau)$, is specified on the original scale. For example, this model assumes the median read counts is associated with covariates on the linear scale. As count/relative abundance data is often skewed, it is more common for such effects to be modelled on a transformed scale. For example, assuming a negative binomial regression model (with offset for library size), the mean relative abundance is associated with covariates with a log link function. This difference does not matter for categorical covariates (e.g. batches), as the contrast is characterized between two levels only. However for continuous covariates the form of the conditional mean function does change with/without transformation. Empirically, evaluation for SBP does indicate good performance ConQuR in the CARDIA study. Still, some discussion regarding this model choice would be helpful.

3. A caveat of the evaluation analyses is that they are purely based on two real-world datasets, where true biological associations are unknown. I think that real-world datasets should be the ultimate standard for such evaluations, but lacking simulation-based analyses does take away from the claim that ConQuR corrected data identifies more “true” biological associations. I do not consider this a vital issue, however it would benefit the publication if some simulation-based analyses can be included, where “true-positives” are known by design.

4. Supp Fig. 3,4: I have two questions here. Using Supp Fig. 3 as the representative:

a. It seems that, visually, ComBat-Seq achieved comparative performance as ConQuR for moderate-to-low prevalence features (0.25-0.5, 0.1-0.25). Is this because ConQuR has difficulties in providing stable estimates with less non-zero read counts, as the authors suggested in Discussion? This is an interesting evaluation, as I’d think most human environment microbes would fall into the moderate-to-low prevalence scenario (< 0.5 prevalence).

b. Why do the right panels (i.e. low prevalence taxa) appear to have fewer observations?

Minor

1. As a general comment, it’d be helpful if line numbers could be included to help referencing specific parts of the manuscript.

2. A few comments regarding Figures 2a/3a:

a. Legends (i.e., what the circles/centroids indicate) should be included in the caption, even though they are explained in the main text.

b. I’m not sure the Bray-Curtis panels should be included for the main figure. Ordination/PERMANOVA based on counts (as opposed to relative abundance) is very rare in practice. They are indeed useful “baseline” evaluations for ConQuR, as the method is designed for counts. Supplementals might be more appropriate for these results.

c. I’m not sure I see the point of the line segments (are they connecting individual samples to the batch centroids)?

3. Page 9, bottom paragraph, the fourth and third lines from last: “Details of data pre-processing and taxonomic assignment are in33”. The reference might be better formatted here (e.g. “Details are published elsewhere33”).

Reviewer #2 (Remarks to the Author):

This study by Ling and colleagues introduces a new methodology to correct for batch effects when combining data from multiple microbiome sequencing studies. The authors employ a two-part model, which captures (1) the probability of presence or absence of a microbiome feature in a sample, and, (2) for features that are present, uses quantile regression to generate corrected read counts for each feature in each sample, normalized to the quantiles of a reference batch. The authors apply this method (ConQuR) on two epidemiological studies, correcting for batch effects within a study and for differences between studies, and provide an R package with functions allowing an end user to run ConQuR batch correction on their data.

Batch effects between microbiome studies are generally acknowledged to be unusually large, relative to most other types of molecular data. Such data also tend to possess unusual statistical properties (as the authors describe well), making it difficult to remove these effects accurately. The use of quantile regression to avoid parametric assumptions on microbiome feature data is thus a new and useful way to correct for batch effects in microbiome studies, and the field has grown enough now that such techniques will be widely applicable for improving meta-analyses. There are some issues, however, that the authors should address in order to strengthen this paper:

Major comments:

--Surprisingly for novel methods development, the manuscript does not include any assessment of how the method performs correcting batch effects of a known size, and identifying associations with key variables of interest of known effect size. A simulation study is relatively standard for such tasks and would greatly strengthen this paper.

As the authors mention, batch effects, if not properly accounted for, can mask true associations between the microbiome and outcomes, or can induce spurious associations. A simulation study would allow the authors to assess A) how well batch effects are removed, B) the increase in accuracy of downstream microbiome-outcome association estimates, and C) the lower limits of abundance and/or prevalence needed to identify associations between outcomes and low-abundance / rare taxa.

We recognize this would probably necessitate using a parametric model to simulate data with known association structure. While this would fall outside the nonparametric nature of the ConQuR

model per se, this is arguably beneficial in that the model would be evaluated on data originating from a different model. The authors would hopefully find that ConQuR performs comparably even when the data is generated under parametric assumptions. At the least, synthetically null conditions can be evaluated by permuting real data, without the need even for simulation.

--The examples used examine the performance of the ConQuR batch correction procedure, while demonstrating the preservation of the relationship between the microbiome and a key variable of interest. However, in both examples, the effect size of the relationship with SBP (example 1) and HIV status (example 2) is quite small, and in the case of the HIV study, is smaller than the post-correction batch effect. It would be helpful to know how the method performs in cases where key variables of interest have a larger association with the microbiome. This could be accomplished via a simulation study, as we suggest above, or by using at least one real dataset with a clearer, less exploratory microbiome association (e.g. body site, antibiotics, IBD, CRC, etc.)

--The main results of the paper come from ConQuR performed using the optional fine-tuning procedures. One of the model fitting types that can be selected for one or more subsets of taxa is the "simple quantile-quantile matching" model, which includes no key variables or covariates. As the authors mention, "ConQuR assumes that for each microorganism, samples share the same conditional distribution if they have identical intrinsic characteristics (with the same values of key variables and important covariates, e.g., clinical, demographic, genetic, and other features), regardless of in which batch they were processed." Consequently, if the ConQuR model does not include any covariates or key variables, then it is assumed that any difference in the distribution of features across batches is due entirely to batch effects. This has the potential to mask important associations or create spurious ones, if variation due to differences in intrinsic characteristics is attributed to batch effects and is therefore removed.

To resolve this, I would 1) ensure that the evaluation procedures used with fine-tuning do not unintentionally "overfit" the tuned parameters (which would result in the ratio of biological covariate to corrected batch covariate being disproportionately high), and 2) provide more evaluations that focus only on quantile-quantile matching (which is similar to the previously published PMID 29684016).

--Following the previous point, the authors do not report which model was selected by the fine-tuning procedure for each subset of taxa in the data; these results should be reported for the main analyses of the paper.

--Also relatedly, it's somewhat surprising that no comparisons with any other microbiome methods (e.g. PMID 29684016, bioRxiv 2020.08.31.261214) are included? Including ComBat is an excellent baseline, but as the authors point out, it was not developed for data with microbiome-like properties.

-- Alternatively / additionally, in the same vein: when random forests are used to quantify the predictive quality of the outputs, it would be helpful to include another model applied to the original dataset, but one that uses batch ID as a covariate. This would allow the authors to quantify the benefit of using ConQuR relative to what is perhaps the simplest way to account for batch structure (that is, beyond ignoring it entirely).

-- The authors repeatedly state that other correction methods fail to allow for visualization. These statements should be made more specific about what type of visualization other methods supposedly don't allow. Any alternative method will allow for visualization to some extent as long as there is some numerical output that could be plotted. I think the authors might mean that other methods do not (straightforwardly) output a corrected abundance table, in addition to hypothesis tests, although even this is not true for most of the alternative methods already mentioned above?

--There is an R package available for easy implementation of the ConQuR method, which is great. However, the accompanying vignette is lacking some detail in explaining the steps and how to interpret the results. Both the help documentation of individual functions and the vignette should reiterate more of the explanatory content from the paper to ensure that less-savvy users can still apply the method.

Moderate comments:

--The name "fine-tuning" could be more descriptive of the two sets of processes it includes. It might be better to simply refer to each step separately: "reference batch selection" and "quantile regression method selection".

--The terms "raw count scale" and "relative abundance scale" should be more explicitly described.

-- In the vignette, setting option(warn = -1) is probably a bad idea. The authors should either fix the source of the warning(s) or explain the (hopefully benign) cause to the user.

-- It would be helpful to hear from the authors at some point on the limit of the number of additional covariates that can feasibly be included before the outputs become unstable when using an input dataset of typical size.

-- It would be interesting to see the authors describe what would happen when ConQuR is applied to data from multiple separate studies each of which have their own within-study batch effects.

-- It would be nice for the authors to show that the corrections are stable against small random perturbations in the input dataset.

-- In the section using linear regression to look for associations between taxa and SBP, it seems unusual that the additional covariates do not include age (only gender and race). It would be helpful to hear the author's motivation for this choice.

-- It would be nice to get uncertainty estimates on the output counts that could then be propagated to downstream association analysis. It seems likely that the count of 1500 that gets corrected to 700 could plausibly also have been corrected to 699 or 701. Could it have been plausibly corrected to 600 or 800? What are the limits of the error bars?

Minor comments:

--On page 10, there's a sentence that needs better wording: "Correcting such data would be more challenging than a relatively homogeneous one like the CARDIA data."

--On page 18, "Not that though only local optima..." should presumably read "Note that though only local optima..."

-- A few modifier words were a bit too emphatic e.g. "direly", "grand".

--The Editorial Policy Checklist requires that "Box-plot elements are defined (e.g. center line, median; box limits, upper and lower quartiles; whiskers, 1.5x interquartile range; points, outliers)"; this is not the case for e.g. Figure 2.

-- In the PCoA plots (Figures 2a and 3a), it might be helpful to have the color legend off to the side (particularly if there are informative identifiers like study names that a user might want to include on the plot).

-- Some of the section titles in the Methods section are vague or poorly worded, like "Trying over choices of reference batch"

--Please clarify the box in Figure 1 that says "Additional Input: Characteristics of Each Sample"; it isn't clear what information is added at this step

-- The Code Availability section only mentions Macintosh or Windows availability, but it seems to run on Linux as well.

-- Figure 3b could be made simpler by putting all three curves on the same panel with different colors.

Reviewer #3 (Remarks to the Author):

The manuscript by Wodan Ling et al. presents a strategy, named Conditional Quantile Regression (ConQuR), to remove batch effects in microbiome data. The proposed methodology is based on a two-part quantile regression model and validated experimentally on two real microbiome data sets.

The topic involved in the paper is suitable for publication in Nature Communications. Overall, the proposed methodology is interesting for the microbiome community since removing of batch effects in microbiome data is a quite understudied topic albeit their importance in the field.

However, I have some comments before a possible publication of the paper:

1. From a very first reading of the manuscript, it may not be very clear if the focus of the methodology is on 16S or shotgun data. Methodological details and experimental validation make more clear that the focus is on 16S data only. Please clarify better this aspect in the text.

2. Following the previous point, if you think instead that the proposed methodology may be also suitable for shotgun data, this should be properly validated in the experimental part.
3. I think the paper lacks a comparison with existing solutions. They are cited in the introduction, but not compared in the validation part of the manuscript. As written by the authors, existing solution may be "only used for batch adjustment in association testing". This is an interesting application anyhow, and it would be interesting how the proposed solution works in comparison with the existing ones.
4. I think that most of the evaluation is discussed from a qualitative point of view only. Could you add more objective evaluation metrics, maybe also considering some simulated datasets?
5. Does the performance of the proposed method depend from the number of studies in your data? Eventually, it would be interesting to see a sensitivity analysis to them.
6. Following the previous point, do you think that other factors can influence the performance of the method? For example the number of sequences? This should be discussed and analyzed in more depth.

Point-to-point responses to referees' comments on NCOMMS-21-35270

We appreciate the comments and opportunity to improve our manuscript. We list all the comments from the referees in italic font and provide our responses below. In the manuscript, the changes corresponding to the comments are in blue.

Reviewer #1 (Remarks to the Author):

In Ling et al., the authors proposed the quantile regression-based method ConQuR, for batch-correction of microbiome count observations. The method is composed of two components targeting the presence/absence and the non-zero part of microbiome count distributions. Additional considerations include model tuning for the reference batch, and a piecewise estimation strategy to prevent instability at the left tail of the non-zero distribution. The authors showcase the method's performance, by applying it to two real-world datasets: one composed of one study of different sequencing batches, the other composed of different cohorts with more prominent "batch" effects. In either case ConQuR demonstrated excellent performance in both reducing overall community profile batch effects, as well as enhancing biological signals post batch correction. It also outperformed existing RNA-Seq batch correction method (ComBat).

I find the proposed quantile regression method elegantly designed, and is well-suited for microbiome data batch correction. The details of the method are clearly explained and easy to follow. The real-world dataset results to showcase ConQuR's performance are convincing and well-presented. Still, I do have a few major-to-moderate comments. These mostly concern the model's design philosophy/interpretations.

Major

1. I wonder about the appropriateness of omitting library size in the model design. The authors suggest that library size effect can be considered as part of the batch-specific effect. This argument applies to sequencing runs as batches, but not necessarily to experimental batches, or study effects. More importantly, removing library size in modelling count data might lead to misleading interpretations. Imagine the toy case of two batches, no other covariate effects. In Batch 1 microbe A has on average 1000 reads out of 3000 total. In Batch 2 microbe A has ~ 2000 reads out of 10000 total. In this example, the count-based batch effect in Batch 2 vs. 1 is positive, whereas the batch effect in relative abundance in Batch 2 vs. 1 is negative.

By extending this toy case to include also biological variables, one can come up with (perhaps contrived) cases where, without modelling for library size, the proposed method might mistakenly estimate both biological and batch signals. The upshot being, for sequencing-based microbiome observations, the per-sample library size is considered an artificial technical signal, and thus often explicitly accounted for in modelling with e.g. offset terms or as a covariate, to avoid confounding effects.

Thus, I wonder why the authors decided not to consider this variable in their modelling. It was pointed out in their evaluations (Figures 2/3) that ConQuR performed better in count-based than for relative abundance-based evaluations. Maybe accounting for library size can help improve this? On the flip note, the evaluations for e.g. G-UniFrac and prediction modelling still showcased ConQuR's superiority over ComBat, so empirically ignoring library size seemed

acceptable for these two datasets.

We thank the reviewer for this thoughtful question. The reviewer has raised an interesting point. ConQuR does indeed comprehensively remove all variability associated with batch, including potential variability in the library size. Including the library size in the model could, in principle, preserve the effect of library size if there is a scientific rationale leading one to consider library size as an important quantity. Consequently, we now also consider a version of ConQuR denoted ConQuR-libsiz. It incorporates the standardized library size as a covariate in the logistic model and includes the library size as an offset in the quantile model. Intrinsically, ConQuR-libsiz models the (logarithm of) relative abundance. Note that the final output of ConQuR-libsiz is still taxonomic read count, as we multiple the batch-free relative abundance with library size to be the final output, similar to what the existing methods do (e.g., MMUPHin).

Evaluated on the simulated data and 3 real datasets, in terms of the effectiveness of removing batch effect, preserving the key variable's signal in both explanatory and predictive metrics, and facilitating valid (controlling false positives) and relatively powerful subsequent individual-taxon association analysis, ConQuR-libsiz demonstrates comparable or improved performance than the existing approaches. However, ConQuR-libsiz is not as good as ConQuR in reducing overall batch effect, while it does allow for consideration of the library size.

To sum up, we feel that in common usage of the method, library size effects are often not of interest and are therefore reasonable to include in the batch correction. Combined with our numerical results, this leads us to generally suggest using ConQuR rather than ConQuR-libsiz unless there is a compelling scientific reason for consideration of library size.

We provide this recommendation and rationale on page 15 of the manuscript (lines 404-410).

2. For the presence-absence logistic modelling, the same library size concern applies. But additionally, I wonder about the appropriateness of converting non-zero counts to zero values during batch correction, as this is a potential outcome of ConQuR (Figure 1b Sample B). Imagine the application case of correcting for sequencing batch effects. For a microbe to have substantial non-zero counts in certain batches, one would reason that that such observations indicate actual biological presence of the microbe, and are not mere sequencing technical effects. Thus, correcting some of these non-zero counts to zero values seems difficult to justify to me. The counterpart – imputing zero values to non-zero values seems more acceptable, as the microbe might not have been detected in certain batches due to e.g. under-detection.

We thank the reviewer for this question. The reviewer has, again, raised an interesting philosophical and interpretive question. We generally agree that setting a non-zero to zero is conceptually odd since we know the microbe is present. However, this is not very different from data coarsening, which is commonly done when we discretize biomarker values, or even rarefying: in each of these cases, we accept the trade-off of ensuring data homogeneity at the cost of increased measurement error.

It may be helpful philosophically to understand the introduced zeros as sampling zeros rather than structural zeros. In microbiome studies, there is essentially no way to differentiate between structural zeros and sampling zeros, so we are effectively presuming that the differences in rate

of microbe presence between batches is primarily due to a higher rate of sampling zeros (not structural zeros) in the sparser batches.

In this way, ConQuR aims to align the other batches' distributions, including the presence-absence likelihood, to the reference batch. It cannot actually recover the true abundances, which is beyond the scope of any batch correction approach. Instead, it simply tries to remove differences between batches. Therefore, even if one of the microbes is missing in the reference batch due to a technical problem, the other batches are matched to the degenerate distribution concentrated at zero. In this regard, ConQuR seems rather brute force, removing batch variation as much as possible relative to a reference batch. However, numerical results (simulation and 3 real datasets) show that it preserves or even amplifies the key biological signals in the data.

Overall, we agree that this is not entirely satisfying, since ideally one would like to recover the “truth”. However, that task is likely asking too much of the limited data that one has and represents a philosophically and operationally different objective than our strategy. We emphasize that other batch correction approaches also suffer from the same logical disconnect, and ConQuR is less extreme than alternative approaches due to its non-parametric nature.

Moderate

1. First, I'd like to complement the authors on their considerations regarding model tuning, and the three-piece estimation strategy. It showed great care in designing a method that's well-tailored to microbiome data applications.

We thank the reviewer for the comment!

2. It seems that the regression coefficients for non-zero counts, $\alpha(\tau)$, is specified on the original scale. For example, this model assumes the median read counts is associated with covariates on the linear scale. As count/relative abundance data is often skewed, it is more common for such effects to be modelled on a transformed scale. For example, assuming a negative binomial regression model (with offset for library size), the mean relative abundance is associated with covariates with a log link function. This difference does not matter for categorical covariates (e.g. batches), as the contrast is characterized between two levels only. However for continuous covariates the form of the conditional mean function does change with/without transformation. Empirically, evaluation for SBP does indicate good performance ConQuR in the CARDIA study. Still, some discussion regarding this model choice would be helpful.

We thank the reviewer for this comment. Although as the reviewer noted, transformations will affect the conditional mean function, because ConQuR is a quantile-based model, its performance is unchanged under monotone transformations such as the log transformation. Mathematically, the model of ConQuR enjoys equivariance to monotone transformations [1], i.e., $Q_{h(Y)}(\tau) = h(Q_Y(\tau))$, where h is some monotone function.

3. A caveat of the evaluation analyses is that they are purely based on two real-world datasets, where true biological associations are unknown. I think that real-world datasets should be the ultimate standard for such evaluations, but lacking simulation-based analyses does take away from the claim that ConQuR corrected data identifies more “true” biological associations. I do not consider this a vital issue, however it would benefit the publication if some simulation-based analyses can be included, where “true-positives” are known by design.

We thank the reviewer for this helpful suggestion. We have now conducted a simulation study based on a real vaginal study, generating simulated data from a Dirichlet distribution with parameters estimated from the observed taxonomic counts in the starting data (mimicking the simulation workflow of ALDEx2 [2]). We simulated two conditions and two batches, which confounded each other, and considered six scenarios, including Null (no batch effects), Condition Effect > Batch Effect, and Condition Effect < Batch Effect, modified by whether the between-batch library size variability is part of batch effects. A subset of taxa were set to be differentially abundant between the two conditions, while batch affected the entire microbial profiles.

We evaluated ConQuR on the simulated data from 3 perspectives: (1) how well the batch effects are removed and condition effects are preserved, (2) the ability of corrected read counts to predict Condition, and (3) the false discovery rate (FDR) and sensitivity of subsequent individual-taxon association analysis for Condition. In terms of the effectiveness of removing batch effects, preserving the key variable’s signals in both explanatory and predictive metrics, and facilitating valid (controlling false positives) and relatively powerful subsequent individual-taxon association analysis, ConQuR demonstrates noticeably improved performance compared to the existing approaches. ConQuR-libsizes seems inferior to ConQuR in some cases but demonstrates similar or improved performance compared to the existing approaches.

The simulation study and results are described in the manuscript on pages 6-9 (lines 152-230).

*4. Supp Fig. 3,4: I have two questions here. Using Supp Fig. 3 as the representative:
a. It seems that, visually, ComBat-Seq achieved comparative performance as ConQuR for moderate-to-low prevalence features (0.25-0.5, 0.1-0.25). Is this because ConQuR has difficulties in providing stable estimates with less non-zero read counts, as the authors suggested in Discussion? This is an interesting evaluation, as I’d think most human environment microbes would fall into the moderate-to-low prevalence scenario (< 0.5 prevalence).
b. Why do the right panels (i.e. low prevalence taxa) appear to have fewer observations?*

a. We thank the reviewer for the question. The reviewer is correct that ConQuR tends to do much better for common taxa, and its advantage compared to ComBat-Seq in rare taxa is not as apparent. Indeed, there are usually more rare taxa than common taxa in real microbiome data. However, at the same time, the influence of rare taxa is usually smaller, i.e., they explain much less variability in the data. Therefore, ConQuR’s overall advantage is still pronounced as it does a much-improved correction on the moderate-to-common frequency taxa and a comparable correction on the low frequency taxa. ComBat-Seq assumes that the read counts follow a negative binomial distribution, which can be strongly violated for common taxa. However, for low frequency taxa, this distributional assumption is less likely to be violated empirically: in the extreme situation where we observe a singleton, there is no way to discern

whether that taxon follows any specific distribution. In such a setting, ConQuR would only do as well as any other approach.

We further emphasize that the concept of rarity is relative and dependent on sample size. If a taxon has a frequency 1/100, then it would be rare (if observed at all) if the sample size is only 100. But if the sample size is 10,000 (as in some emerging large-scale studies), we would expect the taxon to be observed in 100 individuals, in which case we would again expect ConQuR to do much better than other approaches.

b. We thank the reviewer for pointing this out and apologize for not labeling the figure more clearly. Because most samples have zero counts for those rare taxa, they do not appear on the PCoA plots that only contain the rare taxa. We have added this information in the relevant figure legends (now Supp Fig. 5, 6, and 7).

Minor

1. As a general comment, it'd be helpful if line numbers could be included to help referencing specific parts of the manuscript.

Line numbers have been added.

2. A few comments regarding Figures 2a/3a:

a. Legends (i.e., what the circles/centroids indicate) should be included in the caption, even though they are explained in the main text.

b. I'm not sure the Bray-Curtis panels should be included for the main figure.

Ordination/PERMANOVA based on counts (as opposed to relative abundance) is very rare in practice. They are indeed useful "baseline" evaluations for ConQuR, as the method is designed for counts. Supplementals might be more appropriate for these results.

c. I'm not sure I see the point of the line segments (are they connecting individual samples to the batch centroids)?

a. The explanation has been added in legends of Fig. 3a, 4a, 5a, and those relevant in Supp.

b. Because ConQuR is designed for taxonomic read counts and aims to remove all relevant variation, including the between-batch library size variability, Bray-Curtis is an informative and convincing dissimilarity to present. Aitchison and GUniFrac dissimilarities are also present to complete the comprehensive evaluation.

c. Yes, the line segments connect the sample points to the batch centroid.

3. Page 9, bottom paragraph, the fourth and third lines from last: "Details of data pre-processing and taxonomic assignment are in33". The reference might be better formatted here (e.g. "Details are published elsewhere33").

The reference has been changed accordingly.

Reviewer #2 (Remarks to the Author):

This study by Ling and colleagues introduces a new methodology to correct for batch effects when combining data from multiple microbiome sequencing studies. The authors employ a two-part model, which captures (1) the probability of presence or absence of a microbiome feature in a sample, and, (2) for features that are present, uses quantile regression to generate corrected read counts for each feature in each sample, normalized to the quantiles of a reference batch. The authors apply this method (ConQuR) on two epidemiological studies, correcting for batch effects within a study and for differences between studies, and provide an R package with functions allowing an end user to run ConQuR batch correction on their data.

Batch effects between microbiome studies are generally acknowledged to be unusually large, relative to most other types of molecular data. Such data also tend to possess unusual statistical properties (as the authors describe well), making it difficult to remove these effects accurately. The use of quantile regression to avoid parametric assumptions on microbiome feature data is thus a new and useful way to correct for batch effects in microbiome studies, and the field has grown enough now that such techniques will be widely applicable for improving meta-analyses. There are some issues, however, that the authors should address in order to strengthen this paper:

Major comments:

--Surprisingly for novel methods development, the manuscript does not include any assessment of how the method performs correcting batch effects of a known size, and identifying associations with key variables of interest of known effect size. A simulation study is relatively standard for such tasks and would greatly strengthen this paper.

As the authors mention, batch effects, if not properly accounted for, can mask true associations between the microbiome and outcomes, or can induce spurious associations. A simulation study would allow the authors to assess A) how well batch effects are removed, B) the increase in accuracy of downstream microbiome-outcome association estimates, and C) the lower limits of abundance and/or prevalence needed to identify associations between outcomes and low-abundance / rare taxa.

We recognize this would probably necessitate using a parametric model to simulate data with known association structure. While this would fall outside the non-parametric nature of the ConQuR model per se, this is arguably beneficial in that the model would perforce be evaluated on data originating from a different model. The authors would hopefully find that ConQuR performs comparably even when the data is generated under parametric assumptions. At the least, synthetically null conditions can be evaluated by permuting real data, without the need even for simulation.

We thank the reviewer for the helpful suggestion. Reviewer 1 made a similar suggestion (Moderate, Point 3), and we have described the additional simulation study there. We refer the reviewer to that response for points A) and B).

The reviewer has raised an interesting point in C). We note that “low abundance” or “rarity” is a relative concept and depends on sample size (both overall and within a batch). If a taxon has a frequency 1/100, then it would be rare (if observed at all) if the sample size is only 100. But if

the sample size is 10,000 (as in some emerging large-scale studies), we would expect the taxon to be observed in 100 individuals, which is more than enough to fit ConQuR (a minimum of 30 observations is sufficient to conduct significant statistics). Moreover, the power of association analysis depends on both the effect size and variation (the test statistic is $\frac{\text{effect size}}{\text{variation}}$). Therefore, even for a genuinely rare taxon, both the effect size and variation will be small: it is difficult to tell how the rarity will affect association analysis. To sum up, the lower limit varies depending on the sample size and the signal-noise ratio in the corrected data. In the numerical studies (simulation and 3 real datasets), ConQuR-corrected data shows improved performance in controlling false positives while achieving satisfactory power in the subsequent association analysis than competing methods, although we are unable to provide an exact lower bound of taxon prevalence at this time.

--The examples used examine the performance of the ConQuR batch correction procedure, while demonstrating the preservation of the relationship between the microbiome and a key variable of interest. However, in both examples, the effect size of the relationship with SBP (example 1) and HIV status (example 2) is quite small, and in the case of the HIV study, is smaller than the post-correction batch effect. It would be helpful to know how the method performs in cases where key variables of interest have a larger association with the microbiome. This could be accomplished via a simulation study, as we suggest above, or by using at least one real dataset with a clearer, less exploratory microbiome association (e.g. body site, antibiotics, IBD, CRC, etc.)

We thank the reviewer for this suggestion. We explored the recommended datasets, but unfortunately found in each case that the batch effect dominated the variable of interest. This is because a statistically strong (significant) association does not necessarily indicate a large data variability explained by the variable. For example, in the CRC data, the CRC condition only explains 0.66% of the data variability (PERMANOVA R^2 in Aitchison dissimilarity).

Therefore, we have added a real analysis using the MOUTH (Men and Women Offering Understanding of Throat HPV) study [3]. In this dataset, batch effect and the key variable, cigarette smoking status (CIG), explain comparable proportions of the oral microbiome data variability. Moreover, the key variable CIG is a polytomous variable, completing the stories of CARDIA and HIVRC, in which the key variables are continuous and binary, respectively. Visually and numerically, the MOUTH data does not suffer from serious batch variation. All methods can further mitigate the batch effects, but notably, ConQuR did the best job in unifying the means, dispersions, and higher-order features of the sequencing batches. The additional methods and results may be found on pages 13-15 (lines 352-385).

Moreover, Condition Effect > Batch Effect in Scenarios B and E in the simulation study, where the relative effect size is set by condition fold change (FC) >> batch FC and validated by condition PERMANOVA R^2 > batch PERMANOVA R^2 . In those two scenarios, ConQuR still outperforms other approaches in removing batch effects while preserving condition effects and facilitating valid (controlling false positives) and relatively powerful subsequent individual-taxon association analysis. These results may be found in the revised manuscript on page 8-9 (lines 208-224) and in Supp Fig. 4.

--The main results of the paper come from ConQuR performed using the optional fine-tuning procedures. One of the model fitting types that can be selected for one or more subsets of taxa is the “simple quantile-quantile matching” model, which includes no key variables or covariates. As the authors mention, “ConQuR assumes that for each microorganism, samples share the same conditional distribution if they have identical intrinsic characteristics (with the same values of key variables and important covariates, e.g., clinical, demographic, genetic, and other features), regardless of in which batch they were processed.” Consequently, if the ConQuR model does not include any covariates or key variables, then it is assumed that any difference in the distribution of features across batches is due entirely to batch effects. This has the potential to mask important associations or create spurious ones, if variation due to differences in intrinsic characteristics is attributed to batch effects and is therefore removed.

To resolve this, I would 1) ensure that the evaluation procedures used with fine-tuning do not unintentionally “overfit” the tuned parameters (which would result in the ratio of biological covariate to corrected batch covariate being disproportionately high), and 2) provide more evaluations that focus only on quantile-quantile matching (which is similar to the previously published PMID 29684016).

Although batch correction including the variable of interest is the standard approach (for other omics), we agree with the reviewer that fine-tuning could, in principle, lead to over-correction. However, we conducted extensive simulations to explore this possibility and results show that ConQuR well controls the false positive rate (and is better than the competing methods), suggesting that the bias from including the variable of interest is modest.

For the simple quantile-quantile (QQ) matching fitting strategy of ConQuR, we first note that it is fundamentally different from Percentile [4, PMID 29684016]. Percentile assumes that the control group is homogenous regardless of the batch. Thus, it combines control samples from all batches to be the reference pool. Then for each treatment sample, it finds which percentile of the reference pool it equals. In contrast, the QQ matching fitting strategy does not assume any homogeneity across batches. Next, we note that inclusion and exclusion of covariates in ConQuR lead to comparable results when the covariates are very balanced among batches (e.g., CARDIA data). Tab 1 below shows that QQ matching achieves slightly inferior performance in reducing batch effects and preserving SBP effects. When the covariates are imbalanced among batches (e.g., HIVRC data), the inclusion of covariates in ConQuR notably improves performance (Tab 2 below).

Tab 1. CARDIA data

PERMANOVA R^2	Raw count (Bray-Curtis)		Relative abundance (Aitchison)	
	batch	SBP	batch	SBP
Original	0.0566	0.0037	0.0356	0.0035
ConQuR	0.0010	0.0038	0.0086	0.0038
QQ Matching	0.0012	0.0038	0.0257	0.0036

Tab 2. HIVRC data

PERMANOVA R^2	Raw count (Bray-Curtis)		Relative abundance (Aitchison)	
	study	HIV status	study	HIV status
Original	0.3039	0.0057	0.2737	0.0101
ConQuR	0.0194	0.0059	0.0791	0.0106
QQ Matching	0.0669	0.0068	0.1325	0.0072

--Following the previous point, the authors do not report which model was selected by the fine-tuning procedure for each subset of taxa in the data; these results should be reported for the main analyses of the paper.

We thank the reviewer for the suggestion and apologize for the omission of this information. Tables of fitting strategies selected for taxa with different prevalence are added in Supp. Tab. 4-6.

--Also relatedly, it's somewhat surprising that no comparisons with any other microbiome methods (e.g. PMID 29684016, bioRxiv 2020.08.31.261214) are included? Including ComBat is an excellent baseline, but as the authors point out, it was not developed for data with microbiome-like properties.

We thank the reviewer for the suggestion. We have now included Percentile [4] and MMUPHin [5, bioRxiv 2020.08.31.261214] as competing methods in addition to ComBat-seq. We did not previously consider MMUPHin because it literally makes assumptions about and works on the transformed relative abundance rather than directly on the taxonomic read counts. We did not previously consider Percentile because it is designed for case-control studies, works on relative abundance, and produces “percentiles”.

In the numerical studies (simulation and 3 real datasets), ConQuR demonstrates noticeably improved performance compared to the 3 competing methods.

-- Alternatively / additionally, in the same vein: when random forests are used to quantify the predictive quality of the outputs, it would be helpful to include another model applied to the original dataset, but one that uses batch ID as a covariate. This would allow the authors to quantify the benefit of using ConQuR relative to what is perhaps the simplest way to account for batch structure (that is, beyond ignoring it entirely).

We find the reviewer’s suggestion of including batch ID as a predictor in models of the key variable an interesting idea, particularly for the case in which the primary objective is to estimate the ability of the microbiome to predict the key variable. For this modeling approach to be useful, it would have to be the case that being sequenced in a particular batch somehow predicts an individual’s health outcome (and in a way that doesn’t interact with the microbiome). This is

usually avoided (at least in a single study) by strategic choices of sequencing batches – although in an integrative analysis of multiple studies, different baseline outcome frequency could be removed by such an adjustment.

In the context of ConQuR, this is a somewhat different set of questions than the ones we hope to address (or, at best, a subset of the questions we hope to address). In particular, the goal of ConQuR is that a comprehensive microbiome batch removal procedure will permit multiple types of batch-free analysis: prediction of the key variable/outcome using the microbiome (the analysis presupposed in the question above), evaluated by prediction accuracy; explaining variability in the microbiome using the key variable (and batches), evaluated by PERMANOVA R^2 ; and finally, permitting downstream single-taxon analyses. We apologize for any confusion and have tried to better explain this in the main text (see page 8, lines 196-198 in the revised text).

-- The authors repeatedly state that other correction methods fail to allow for visualization. These statements should be made more specific about what type of visualization other methods supposedly don't allow. Any alternative method will allow for visualization to some extent as long as there is some numerical output that could be plotted. I think the authors might mean that other methods do not (straightforwardly) output a corrected abundance table, in addition to hypothesis tests, although even this is not true for most of the alternative methods already mentioned above?

We apologize for not being clear about this and thank the reviewer for pointing it out. The reviewer is correct in interpreting our statements: we do mean that most existing batch correction methods tailored for microbiome data focus on association testing, such as [6]. Percentile [4] generates batch-free data but is designed for case-control studies. The previous statement about visualization was changed to: “At the same time, the limited work on batch effects correction tailored for microbiome data can only be used for batch adjustment in association testing, or require specific types of controls/spike-ins. These approaches fail to allow other common analytic goals such as visualization or general study designs.” (see page 3, lines 69-72 in the revised text) We then differentiated batch removal from batch adjustment, with the former referring to generating a batch-free dataset and the latter referring to including batch ID in association testing (see page 4, lines 77-80 in the revised text). Also, we changed the Abstract accordingly.

Overall, MMUPHin [5] is the only tailored method that could generate batch-free data in general designs. We did not consider it previously because it works on the transformed relative abundance and corrects the mean and variance, while ConQuR directly focuses on the taxonomic read counts and corrects the entire conditional distributions non-parametrically.

--There is an R package available for easy implementation of the ConQuR method, which is great. However, the accompanying vignette is lacking some detail in explaining the steps and how to interpret the results. Both the help documentation of individual functions and the vignette should reiterate more of the explanatory content from the paper to ensure that less-savvy users can still apply the method.

We thank the reviewer for this suggestion, which will improve the usability of our software. We have now updated the R package and accompanying documentation with more explanations and details.

Moderate comments:

--The name "fine-tuning" could be more descriptive of the two sets of processes it includes. It might be better to simply refer to each step separately: "reference batch selection" and "quantile regression method selection".

We thank the reviewer for this helpful suggestion. The titles in Methods have been changed to "Selection of fitting strategies for common to rare taxa" and "Selection of the reference batch".

--The terms "raw count scale" and "relative abundance scale" should be more explicitly described.

We thank the reviewer for this suggestion and apologize for any confusion with terminology. The two terms are now explained in the Introduction, when introducing MMUPHin: "...only appropriate for certain transformations of relative abundance data (i.e., taxon counts normalized by each sample's library size)".

-- In the vignette, setting option(warn = -1) is probably a bad idea. The authors should either fix the source of the warning(s) or explain the (hopefully benign) cause to the user.

We thank the reviewer for pointing this out. option(warn = -1) was used in the vignette to suppress a benign warning from quantile regression indicating the solution is not unique. Since quantile regression is conducted many times throughout the vignette, it becomes unwieldy and difficult to read if the warnings are not suppressed. We have updated the vignette with a statement in the beginning explaining why such an option is used.

-- It would be helpful to hear from the authors at some point on the limit of the number of additional covariates that can feasibly be included before the outputs become unstable when using an input dataset of typical size.

We thank the reviewer for this question. Generally, we want $n > (p+k-1)$ for the non-zero sub-data (here, n refers to non-zero observations of the microbe being corrected, and p refers to the dimension of additional covariates), where k batches/studies contribute $k-1$ dimensions of covariates. The standard fitting strategies does not correct taxa that are so rare that $n \leq (p+k-1)$, leaving the original data untouched. However, we have penalized fitting strategies to tackle the high-dimensional covariate problem. This is exactly one of the reasons why we proposed the tuning procedure, to use different fitting strategies for taxa with different prevalence.

-- It would be interesting to see the authors describe what would happen when ConQuR is

applied to data from multiple separate studies each of which have their own within-study batch effects.

We thank the reviewer for this question. We don't differentiate between levels of batch/study effects, although as the reviewer noted, the variation is moderate between sequencing batches, while there is much more heterogeneity between studies. The only condition required to use ConQuR is that the sources of variation are known. Therefore, if we know the batch ID within each of the studies and regard the finest-level batch as our batch unit, ConQuR also can align the data across all studies and batches to the chosen reference batch.

-- It would be nice for the authors to show that the corrections are stable against small random perturbations in the input dataset.

The reviewer has raised an interesting question. We added a small perturbation, (-1, 0, 1) with probability (0.25, 0.5, 0.25), to the CARDIA data, and ran ConQuR on the perturbed data. We repeated the procedure 10 times and summarized the batch and SBP PERMANOVA R^2 (in Bray-Curtis and Aitchison dissimilarities) in the corrected data using boxplots. Fig 1 below shows that ConQuR is very stable in producing batch-free taxonomic read counts, on which batch and SBP PERMANOVA R^2 vary minimally.

Fig 1. PERMANOVA R^2 explained by batch ID and SBP in 10 perturbed CARDIA data.

-- In the section using linear regression to look for associations between taxa and SBP, it seems unusual that the additional covariates do not include age (only gender and race). It would be helpful to hear the author's motivation for this choice.

We thank the reviewer for this question. We chose to omit age in order to accommodate ComBat-seq. Its algorithm of generating batch-free counts fails if age is added. This is because taxa that are highly dispersed or with extreme abundances can lead to large estimates of either

the mean or dispersion parameters of negative binomial distribution. As the continuous variable, age, adds even finer segmentation of the data, the estimates are even larger, making the new count generating algorithm run forever and fail to converge. This is stated in Methods, Computation of ConQuR (see page 24, lines 627-634 in the revised text).

-- It would be nice to get uncertainty estimates on the output counts that could then be propagated to downstream association analysis. It seems likely that the count of 1500 that gets corrected to 700 could plausibly also have been corrected to 699 or 701. Could it have been plausibly corrected to 600 or 800? What are the limits of the error bars?

The reviewer has raised an interesting question. The exact form of uncertainty is currently unclear. Theoretically, the uncertainty in the final corrected read count comes from (1) how accurate we estimate the quantile level of the observed read count and (2) how accurate we estimate the batch-free conditional quantile function. Fundamentally, (1) and (2) depend on the uncertainty of estimating a conditional quantile function using the two-part quantile regression model. Therefore, we give the asymptotic results below (based on the notations in the manuscript and supplementary material):

With mild regularity conditions and definitions below:

$$\begin{aligned} D_{1,\theta^Q(\tau)} &= E \left[\pi(\theta^L, X) f_{W|Y>0}(X^T \theta^Q(\tau)|X) X X^T \right], \\ D_0 &= E \{ \pi(\theta^L, X) X X^T \}, \\ D_{1,\theta^L} &= E \left[\pi(\theta^L, X) \{1 - \pi(\theta^L, X)\} X X^T \right], \\ \dot{\theta}^Q(\tau) &= \frac{d \theta^Q(t)}{d t} \Big|_{t=\tau}, \end{aligned}$$

we have

(1) If $\tau < 1 - \pi(\theta^L, X)$, $\hat{Q}_Y(\tau|X)$ is super-efficient, i.e., as $n \rightarrow \infty$,
 $\sqrt{n} (\hat{Q}_Y(\tau|X) - 0) \rightarrow_p 0$.

(2) If $\tau = 1 - \pi(\theta^L, X)$, denote $Q'_Y(0|X, Y > 0)$ as the right derivative, which is well defined because $\theta^Q(\tau)$ is right differentiable at zero. Then as $n \rightarrow \infty$,

$$\sqrt{n} (\hat{Q}_Y(\tau|X) - 0) \rightarrow_d \{1 - \pi(\theta^L, X)\} \sqrt{X^T D_{1,\theta^L}^{-1} X} Q'_Y(0|X, Y > 0) Z_0 I(Z_0 > 0),$$

where $Z_0 \sim N(0,1)$.

(3) If $\tau > 1 - \pi(\theta^L, X)$, as $n \rightarrow \infty$,
 $\sqrt{n} (\hat{Q}_Y(\tau|X) - Q_Y(\tau|X)) \rightarrow_d N(0, \Sigma_1 + \Sigma_2)$,

where

$$\begin{aligned} \Sigma_1 &= \Gamma(\tau; X, \theta^L) \{1 - \Gamma(\tau; X, \theta^L)\} X^T D_{1,\theta^Q \circ \Gamma(\tau; X, \theta^L)}^{-1} D_0 D_{1,\theta^Q \circ \Gamma(\tau; X, \theta^L)}^{-1} X, \\ \Sigma_2 &= \{1 - \Gamma(\tau; X, \theta^L)\}^2 \{1 - \pi(\theta^L, X)\}^2 X^T D_{1,\theta^L}^{-1} X \cdot X^T \dot{\theta}^Q \circ \Gamma(\tau; X, \theta^L) \dot{\theta}^Q \circ \Gamma(\tau; X, \theta^L)^T X. \end{aligned}$$

With the asymptotic distributions, we can construct any $(1 - \alpha) \times 100\%$ confidence band, e.g., $\alpha=0.05$, to cover the estimate of the τ th conditional quantile. However, we further emphasize

that this is conditional upon the quantile level τ and a more general theory about the corrected value is beyond the scope of this paper but is something that warrants further theoretical investigation. Regardless, we believe that the empirical evaluations justify the reasonable performance of our strategy.

Minor comments:

--On page 10, there's a sentence that needs better wording: "Correcting such data would be more challenging than a relatively homogeneous one like the CARDIA data."

The sentence has been changed to "Correcting such heterogenous microbiome data is more challenging than correcting the CARDIA data".

--On page 18, "Not that though only local optima..." should presumably read "Note that though only local optima..."

"Not" has been changed to "Note".

-- A few modifier words were a bit too emphatic e.g. "direly", "grand".

Those words have been deleted.

--The Editorial Policy Checklist requires that "Box-plot elements are defined (e.g. center line, median; box limits, upper and lower quartiles; whiskers, 1.5x interquartile range; points, outliers)"; this is not the case for e.g. Figure 2.

We thank the reviewer for pointing this out. The definitions have been added in caption of Fig. 3c, 5c, and those relevant in Supp.

-- In the PCoA plots (Figures 2a and 3a), it might be helpful to have the color legend off to the side (particularly if there are informative identifiers like study names that a user might want to include on the plot).

The color legends have been added in Fig. 3c, 5c, and those relevant in Supp.

-- Some of the section titles in the Methods section are vague or poorly worded, like "Trying over choices of reference batch"

The titles in Methods have been changed to "Selection of fitting strategies for common to rare taxa", "Selection of the reference batch".

--Please clarify the box in Figure 1 that says "Additional Input: Characteristics of Each Sample"; it isn't clear what information is added at this step

Characteristics has been changed to Metadata.

-- The Code Availability section only mentions Macintosh or Windows availability, but it seems to run on Linux as well.

We thank the reviewer for noticing this. Linux has been added to Code Availability.

-- Figure 3b could be made simpler by putting all three curves on the same panel with different colors.

ROC-AUC curves have now been plotted together in Fig. 4c.

Reviewer #3 (Remarks to the Author):

The manuscript by Wodan Ling et al. presents a strategy, named Conditional Quantile Regression (ConQuR), to remove batch effects in microbiome data. The proposed methodology is based on a two-part quantile regression model and validated experimentally on two real microbiome data sets.

The topic involved in the paper is suitable for publication in Nature Communications. Overall, the proposed methodology is interesting for the microbiome community since removing of batch effects in microbiome data is a quite understudied topic albeit their importance in the field.

However, I have some comments before a possible publication of the paper:

1. From a very first reading of the manuscript, it may not be very clear if the focus of the methodology is on 16S or shotgun data. Methodological details and experimental validation make more clear that the focus is on 16S data only. Please clarify better this aspect in the text.

We apologize for not making this clear. We have now clarified this in the Discussion, page 16, lines 425-427: "Note that we only examined 16S rRNA data in the paper. However, methodologically, ConQuR can also be extended and applied to full genome data, which is one of our future directions."

2. Following the previous point, if you think instead that the proposed methodology may be also suitable for shotgun data, this should be properly validated in the experimental part.

We very much agree with the reviewer and add shotgun applications as one future direction in Discussion: "Note that we only examined 16S rRNA data in the paper. However,

methodologically, ConQuR can also be extended and applied to full genome data, which is one of our future directions.”

3. I think the paper lacks a comparison with existing solutions. They are cited in the introduction, but not compared in the validation part of the manuscript. As written by the authors, existing solution may be "only used for batch adjustment in association testing". This is an interesting application anyhow, and it would be interesting how the proposed solution works in comparison with the existing ones.

We thank the reviewer for the suggestion. In all the numerical studies, we included Percentile [4] and MMUPHin [5] as competing methods in addition to ComBat-seq. ConQuR demonstrates noticeably improved performance than the 3 competing methods.

Those only workable in associating testing, such as [6], are difficult to compare in the context of ConQuR because our major comparison tools of visualization, PERMANOVA R^2 , and predictive metrics cannot be applied.

4. I think that most of the evaluation is discussed from a qualitative point of view only. Could you add more objective evaluation metrics, maybe also considering some simulated datasets?

We thank the reviewer for this helpful suggestion and have added a simulation study, further described in our response to Reviewer 1, Moderate, Point 3 above.

5. Does the performance of the proposed method depend from the number of studies in your data? Eventually, it would be interesting to see a sensitivity analysis to them.

We agree with the reviewer that the number of batches/studies will affect the performance of all batch removal methods, including ConQuR. In addition to the number, the size of batches/studies is also important. For example, having many small batches (resulting in limited information within a batch) results in worse performance for all methods, but if each batch is large, having many batches does not necessarily hinder the correction.

For ConQuR in particular, its two-part quantile regression model regards batches/studies as dummy variables. Generally, we want $n > (p+k-1)$ for the non-zero sub-data (here, n refers to non-zero observations of the microbe being corrected, and p refers to the dimension of additional covariates), where k batches/studies contribute $k-1$ dimensions of covariates. The standard fitting strategies does not correct taxa that are so rare that $n \leq (p+k-1)$, leaving the original data untouched. But we have penalized fitting strategies to tackle the high-dimensional covariate problem. This is one of the reasons why we proposed the tuning procedure, to use different fitting strategies for taxa with different prevalence. The numerical studies (simulation and 3 real datasets) include 2, 6, and 10 batches/studies, which all demonstrate noticeably improved performance of ConQuR as compared to the existing methods.

6. Following the previous point, do you think that other factors can influence the performance of

the method? For example the number of sequences? This should be discussed and analyzed in more depth.

This is, again, an interesting question that goes beyond just ConQuR. In general, there are many factors affecting performance of batch removal approaches. The main factors include the magnitude of the batch effects and the magnitude of the variable of interest which we now thoroughly investigate through simulation studies. Other factors are more difficult to evaluate, but center largely on the amount of data that can be used to estimate the batch effects (regardless of the method used). For most methods, this will involve the number of positive (non-zero) measurements in total and across batch. Consequently, small batches (resulting in limited information within a batch) and too few sequences (resulting in low-quality information overall) result in worse performance for all methods. We have added a note about this in the Discussion, (page 16-17, lines 435-438)

References

[1] Koenker, R. Econometric Society Monographs: Quantile Regression. New York: Cambridge University (2005).

[2] Fernandes, A.D. et al. Unifying the analysis of high-throughput sequencing datasets: characterizing RNA-seq, 16S rRNA gene sequencing and selective growth experiments by compositional data analysis. *Microbiome* 2, 1-13 (2014).

[3] Zhang, Y. et al. Oral HPV associated with differences in oral microbiota beta diversity and microbiota abundance. *The Journal of infectious diseases* (2022).

[4] Gibbons, S.M., Duvall, C. & Alm, E.J. Correcting for batch effects in case-control microbiome studies. *PLoS computational biology* 14, e1006102 (2018).

[5] Ma, S. et al. Population Structure Discovery in Meta-Analyzed Microbial Communities and Inflammatory Bowel Disease. *bioRxiv* (2020).

[6] Dai, Z., Wong, S.H., Yu, J. & Wei, Y. Batch effects correction for microbiome data with Dirichlet-multinomial regression. *Bioinformatics* 35, 807-814 (2019).

REVIEWER COMMENTS

Reviewer #1 (Remarks to the Author):

I appreciate the authors' scope of work in responding to the reviewers' comments. For the most part, I consider my questions appropriately and satisfactorily addressed. I do have two additional comments, though. I consider the first one more important than the second one.

1. I'm not convinced by the response to my moderate comment #2 (the scale of the regression coefficient, α). It is unclear why the conditional quantile function does not change with monotone functions. Using the conditional median as an example, there are two options:

a. $Q_W(0.5 | X, Y > 0) = Z * \alpha(0.5) + B * \beta(0.5)$

b. $Q_{\{\log(W)\}}(0.5 | X, Y > 0) = Z * \alpha(0.5) + B * \beta(0.5)$

Maybe I'm mistaken, but I think a. assumes that median counts increases by $\alpha(0.5)$ with every unit of change in Z , whereas b. assumes that median counts grows by $\exp(\alpha(0.5))$ fold with every unit of change in Z . These are different models. It might be argued that the author's choice (a.) is more appropriate, but to claim they are equivalent seems incorrect.

Also, confusingly, in their expanded method accounting for library size (lines 544-545), they indeed adopted option b. What is the motivation for this difference between the two versions?

Last, as a minor point, I think the libsize_i term in equation (2) (line 545) should be $\log(\text{libsize}_i)$.

2. I appreciate the authors' discussion on the model philosophy for batch-correcting non-zero counts to zeros (my major comment #2). I particularly liked the idea of interpreting the corrected zeros as sampling zeros, as the concept does make sense and is not often discussed in microbiome literature. This is not vital, but I'd love to see some of these discussions included in the main text, maybe in Discussion or Methods.

Reviewer #3 (Remarks to the Author):

I feel the authors have answered positively to my previous comments and modified the manuscript according to them.

We appreciate the comments and opportunity to improve our manuscript. We list all the comments from the referees in italic font and provide our responses below. In the manuscript, the changes corresponding to the comments are in blue.

Reviewer #1 (Remarks to the Author):

I appreciate the authors' scope of work in responding to the reviewers' comments. For the most part, I consider my questions appropriately and satisfactorily addressed. I do have two additional comments, though. I consider the first one more important than the second one.

1. I'm not convinced by the response to my moderate comment #2 (the scale of the regression coefficient, α). It is unclear why the conditional quantile function does not change with monotone functions. Using the conditional median as an example, there are two options:

*a. $Q_{W(0.5 | X, Y > 0)} = Z * \alpha(0.5) + B * \beta(0.5)$*

*b. $Q_{\{\log(W)\}(0.5 | X, Y > 0)} = Z * \alpha(0.5) + B * \beta(0.5)$*

Maybe I'm mistaken, but I think a. assumes that median counts increases by $\alpha(0.5)$ with every unit of change in Z , whereas b. assumes that median counts grows by $\exp(\alpha(0.5))$ fold with every unit of change in Z . These are different models. It might be argued that the author's choice (a.) is more appropriate, but to claim they are equivalent seems incorrect. Also, confusingly, in their expanded method accounting for library size (lines 544-545), they indeed adopted option b. What is the motivation for this difference between the two versions? Last, as a minor point, I think the libsize_i term in equation (2) (line 545) should be $\log(\text{libsize}_i)$.

We thank the reviewer for pointing out our mistakes and apologize for the lack of clarity in our manuscript.

We agree with the reviewer that the interpretations of $\alpha(\tau)$ are different in Model (a) and Model (b) – a unit change of Z is associated with an increase in the median of $W|Y > 0$ by $\alpha(\tau)$ and $\exp(\alpha(\tau))$, respectively. To hopefully clarify further the meaning of our previous statement, we would like to comment on two properties of quantile regression. First, quantile regression is a rank / order statistical method, i.e., we order the observed W given a realization of (Z, B) (if possible), count to half of the observed W and pick the value as the conditional median. Therefore, the median of $\log(W)$ is equivalent to \log of the median of W , since \log transformation does not change the order of the observed W . This is what we meant by saying the two models are “equivalent”. More accurately, it should be that “the quantile models enjoy equivariance to monotone transformations” [1]. With a monotone transformation on W , such as \log , Model (a) and Model (b) will lead to the same inference and conclusion about W , even though the estimates of $\alpha(\tau)$ will be different and have distinct interpretations, as the reviewer pointed out. Second, since quantile regression is a nonparametric method releasing the Gaussian assumption of ordinary linear regression, it can robustly handle outcomes from any distributions, including the skewed distributions of the jittered taxonomic count of microbiome.

In contrast, the mean of $\log(W)$ is not equivalent to \log of the mean of W due to Jensen's inequality, and the \log transformation is necessary to change the skewed distributions to bell-shaped distributions, so as to meet the Gaussian assumption for ordinary linear regressions. As such, due to the equivariance property and robustness of quantile regression, it is valid and

appropriate to model either W or $\log(W)$. We take Model (a) in ConQuR because it is more straightforward in its interpretation: $\alpha(\tau)$ is the expected increase in quantiles of W with every unit of change in Z . The estimation process is also easier as we simply follow Machado and Silva (2005) [2] (the implementation is by the *dither* function of the *quantreg* package [3]) and there is no need to exponentiate the estimate to recover W .

In ConQuR-libsize, to correct the relative abundance instead of the taxonomic count and to include library size as an offset in the quantile regression model, we follow a standard technique and model $\log(Y)$. This change is not for technical purposes, as modeling either Y or $\log(Y)$ is equivariant as described above, but instead to follow the commonly-used interpretation in parametric modeling. Parametrically, we usually use Poisson regression with an offset to model count data, and the estimated coefficients can be interpreted as changes in the log of the rate of an event. Similarly, when we use quantile regressions for the (log of) taxonomic count with library size as an offset, the estimated coefficients are changes in the log of the relative abundance. This standard technique of quantile regression for modeling count with an offset, though it is not officially stated in a paper, can be found in the *rq.counts* function of the *Qtools* package [4].

We thank the reviewer again for pointing out the typo in Equation (2). We changed $libsize_i$ to $\log(libsize_i)$ (Line 554).

2. I appreciate the authors' discussion on the model philosophy for batch-correcting non-zero counts to zeros (my major comment #2). I particularly liked the idea of interpreting the corrected zeros as sampling zeros, as the concept does make sense and is not often discussed in microbiome literature. This is not vital, but I'd love to see some of these discussions included in the main text, maybe in Discussion or Methods.

We thank the reviewer for this suggestion. We included the model philosophy in Methods (Line 527-536).

Reviewer #3 (Remarks to the Author):

I feel the authors have answered positively to my previous comments and modified the manuscript according to them.

We thank the reviewer for this comment!

References

[1] Koenker, R. Econometric Society Monographs: Quantile Regression. New York: Cambridge University (2005).

[2] Machado, J. A. F., & Silva, J. S. Quantiles for counts. Journal of the American Statistical Association, 100(472), 1226-1237 (2005).

[3] Koenker, R., Portnoy, S., Ng, P. T., Zeileis, A., Grosjean, P., & Ripley, B. D. Package 'quantreg'. Cran R-project. org (2018).

[4] Geraci, M. Qtools: A Collection of Models and Tools for Quantile Inference. R J., 8(2), 117 (2016).

REVIEWERS' COMMENTS

Reviewer #1 (Remarks to the Author):

I appreciate the authors' response to my questions. I have no further comments on the manuscript.